# 3D Scene Prompting for Scene-Consistent Camera-Controllable Video Generation

**JoungBin Lee**[1]* **Jaewoo Jung**[1]* **Jisang Han**[1]* **Takuya Narihira**[2] **Kazumi Fukuda**[2]
**Junyoung Seo**[1] **Sunghwan Hong**[3,5] **Yuki Mitsufuji**[2,4]† **Seungryong Kim**[1]†
[1]KAIST AI  [2]Sony AI  [3]ETH Zürich (CVG, PRS)  [4]Sony Group Corporation
[5]ETH AI Center

https://cvlab-kaist.github.io/3DScenePrompt

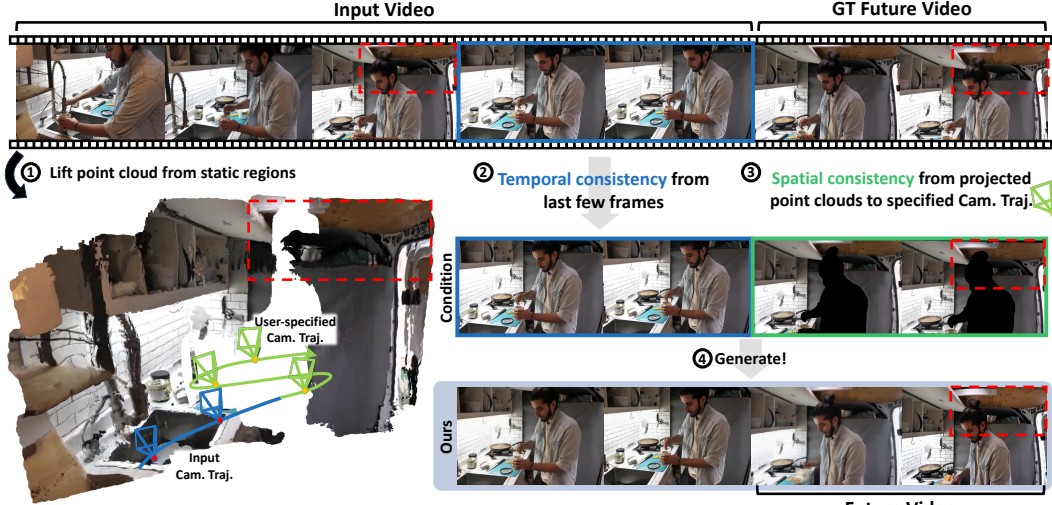

Figure 1: **Teaser.** Our framework generates the next video chunk that follows a user-specified camera trajectory while maintaining scene consistency. Our dual spatio-temporal conditioning jointly leverages the last few frames to ensure temporal continuity and the rendered point cloud to enforce spatial consistency.

## ABSTRACT

We present **3DScenePrompt**, a framework that generates the next video chunk from arbitrary-length input video chunk while supporting *highly complex and precise camera control* and preserving scene consistency. Unlike previous methods conditioned on a single image or a short clip, our approach employs *dual spatio-temporal conditioning* that reformulates context-view referencing across the entire input video. Specifically, we condition on both *temporally* adjacent frames to ensure motion continuity and *spatially* adjacent content to preserve scene consistency, enabled by a *3D scene memory* that exclusively represents the static geometry extracted from the full input sequence. To construct this memory, we leverage *dynamic SLAM* with a newly introduced *dynamic masking strategy* that explicitly separates static scene geometry from moving elements. The resulting static representation can then be projected to arbitrary target viewpoints, providing geometrically consistent warped views that act as strong *3D spatial prompts*, while allowing dynamic regions to evolve naturally from temporal context. This design allows our model to maintain long-range spatial coherence and precise camera control without compromising computational efficiency or motion realism. Extensive experiments demonstrate that our framework significantly outperforms existing methods in scene consistency, camera controllability, and generation quality.

---

*Equal contribution.
†Co-corresponding authors.

# 1 INTRODUCTION

Camera-controllable video generation (He et al., 2024; Wang et al., 2024b; Jin et al., 2025; Hong et al., 2023) aims to synthesize videos following user-specified camera trajectories while maintaining visual coherence and temporal consistency. Recent advances have progressed from generating entirely new videos with controllable viewpoints (Bahmani et al., 2025a) to enabling users to extend a single image or short video clips along desired camera paths (He et al., 2024; Agarwal et al., 2025). Yet these methods share a fundamental limitation: they can only process extremely short conditioning sequences, typically just a few frames, which constrains their ability to understand longer videos and hence fails to preserve the rich scene context present in those longer videos. *What if we could provide a model with arbitrary-length video sequences and generate continuations that not only follow precise camera controls but also maintain scene consistency with the entire input?* Such technology, which we refer to as *scene-consistent camera-controllable video generation*, has immediate applications in film production (Zhang et al., 2025), virtual reality (He et al., 2025b; Hong et al., 2024b; Cho et al., 2024), robotics (Yoon et al., 2025), and synthetic data generation (Knapp & Bohacek, 2025).

Scene-consistent camera-controllable video generation poses three intertwined challenges that must be solved jointly. First, static and dynamic elements must be handled differently: while static scene elements should remain consistent throughout generation, dynamic elements such as moving objects and people should evolve naturally from their most recent states rather than rigidly preserving motions from the distant past. Second, camera control demands understanding the underlying 3D geometry of the scene: the generated content must respect physical constraints, properly handle occlusions, and seamlessly compose dynamic elements onto static geometry, while extrapolating plausible content for previously unobserved regions. Third, these capabilities must be achieved within practical computational constraints, as naive approaches that process all input frames quickly become intractable when the input video sequence is long.

*How can we tackle this challenging task by leveraging existing video generative models?* Our key insight comes from fundamentally rethinking how video models should reference prior content. Current image-to-video (Yang et al., 2024) and video-to-future-video models [1] (Agarwal et al., 2025) achieve realistic generation by conditioning on *temporally adjacent* frames to maintain short-term consistency and motion continuity. However, adjacency in video is not purely temporal—it can also be *spatial*. When generating scene-consistent videos, the frames we synthesize may be spatially adjacent to frames from much earlier in the input sequence, particularly when the camera revisits similar viewpoints or explores nearby regions. This dual nature of adjacency suggests a new conditioning paradigm that leverages both temporal and spatial relationships.

Based on these motivations, we propose **3DScenePrompt**, a novel video generation framework designed for scene-consistent camera-controllable video synthesis. It takes an arbitrary-length video as context and generates the future video that is consistent with the scene geometry of the context video. The key innovation lies in our dual spatio-temporal conditioning strategy: the model conditions on both *temporally* adjacent frames (for motion continuity) and *spatially* adjacent frames (for scene consistency). However, an important consideration for spatial conditioning for our task is that it must provide only the persistent *static* scene structure while excluding *dynamic* content, as directly conditioning on spatially adjacent frames from the past would incorrectly preserve dynamic elements. To enable this without temporal contradictions, we construct a **3D scene memory** that represents exclusively the *static* geometry extracted from the entire input video.

To construct this 3D scene memory from *dynamic* videos, we leverage recent advances in dynamic SLAM frameworks (Zhang et al., 2022; Han et al., 2025a; An et al., 2025; Han et al., 2025b; Zhang et al., 2024; Li et al., 2024) to estimate camera poses and 3D structure from the input video. To extract only the *static* regions from the estimated 3D structure, we introduce a dynamic masking strategy that explicitly separates static elements and moving objects. The static-only 3D representation can then be projected to target viewpoints, yielding geometrically-consistent warped views that serve as *spatial prompts* while allowing dynamic elements to evolve naturally from temporal context alone. Surprisingly, the integration of 3D scene memory provides an additional benefit: the geometrically-consistent warped views provide rich visual references that significantly reduce

---

[1]Throughout our paper, video-to-future-video models refer to models that are capable of generating the subsequent frames of the given input video (e.g., `cosmos-predict2` (Agarwal et al., 2025).

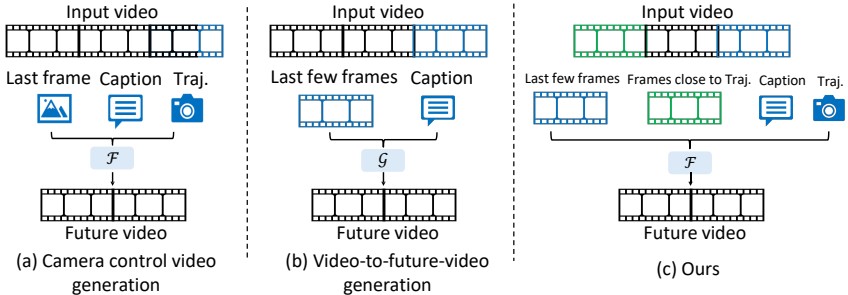

Figure 2: **Comparison to existing architectures.** (a) Camera-controllable video generation methods condition on a single frame and camera trajectory (Wang et al., 2024b; He et al., 2024; Jin et al., 2025). (b) The video-to-future-video generation method (Agarwal et al., 2025) leverages the last few frames of the input video to preserve temporal continuity in future video generation; however, it fails to maintain long-term spatial consistency when revisiting previously defined viewpoints that are not visible in the given frames. Unlike these, (c) our approach combines temporal conditioning (last few frames) with spatial conditioning (spatially adjacent frames) to achieve scene-consistent generation with precise camera control.

uncertainty in viewpoint manipulation, enabling precise camera control without any other explicit camera conditioning.

In summary, **3DScenePrompt** enables both accurate camera control and long-range spatial consistency by treating the static scene representation as a persistent spatial prompt that guides generation across arbitrary timescales. Extensive experiments demonstrate that our framework significantly outperforms existing methods in maintaining scene consistency, achieving precise camera control, and generating high-quality videos from arbitrary-length inputs.

## 2 RELATED WORK

**Single-frame conditioned camera-controllable video generation.** Building upon the recent success of video diffusion models (Blattmann et al., 2023; Guo et al., 2023; Yang et al., 2024; Runway, 2024; Kim et al., 2025; Brooks et al., 2024), recent works (He et al., 2024; Wang et al., 2024b; Bahmani et al., 2024) have achieved camera-controllable video generation by introducing additional adapters into U-Net-based video diffusion models that accept camera trajectories. For instance, CameraCtrl and VD3D (Bahmani et al., 2024; He et al., 2024) incorporate spatiotemporal camera embeddings, such as Plücker coordinates, via ControlNet-like mechanisms (Zhang et al., 2023). While these methods enable precise trajectory following, they only condition on single starting images, lacking mechanisms to maintain consistency with extended video context. In contrast, our approach enables leveraging entire video sequences as spatial prompts through 3D memory construction, enabling scene-consistent generation that preserves the rich scene context within arbitrary-length inputs.

**Multi-frame conditioned camera-controllable video generation.** Recently, CameraCtrl2 (He et al., 2025a) and Seaweed-APT2 (Lin et al., 2025b) have proposed to take multiple frames as a condition for camera-controllable video generation. This allows the generated videos to maintain temporal smoothness with the provided frames, enhancing the motion fidelity of the generated videos. However, these methods only consider temporal adjacency, which restricts the model from maintaining scene-consistencies with long videos due to memory constraints. In contrast, we introduce SLAM to process the conditioning video to consider dual spatio-temporal adjacency, enabling the model to maintain scene-consistency with long videos under efficient computation.

**Geometry-grounded video generation.** Recent works (Ren et al., 2025; Yu et al., 2025; Seo et al., 2025) have integrated off-the-shelf geometry estimators Li et al. (2024); Wang et al. (2025) into video generation pipelines to improve geometric accuracy. Gen3C (Ren et al., 2025), for instance, similarly adopts dynamic SLAM to lift videos to 3D representations. However, these methods exclusively address dynamic novel view synthesis—generating new viewpoints within the same temporal window as the input. This constrained setting allows them to simply warp entire scenes

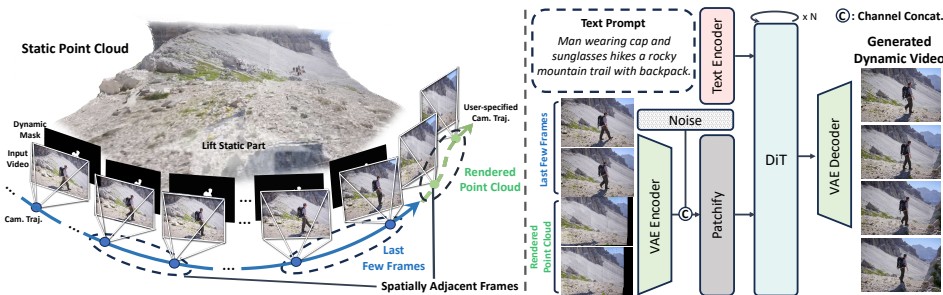

Figure 3: **Overview of 3DScenePrompt framework.** To generate the next chunk video that remains spatially consistent with the input video, we design a dual spatio-temporal conditioning pipeline to extract the most relevant information from the input video. The last few frames are utilized to provide temporal conditioning, ensuring motion continuity between conditioning inputs and the generated frames. In parallel, for the spatial conditioning, we first select the most representative frames from the input sequence, lift their static regions into a 3D point cloud using the dynamic mask, and render it along a user-specified camera trajectory to preserve scene geometry.

without distinguishing static and dynamic elements. Our work fundamentally differs by generating content beyond temporal boundaries, requiring selective masking of dynamic regions during 3D construction—a critical challenge that emerges only when static geometry must persist while dynamics evolve naturally into the future.

**Long-horizon scene-consistent generation.** Various approaches attempt scene-consistent long video generation through different strategies. ReCamMaster (Bai et al., 2025) and TrajectoryCrafter (Yu et al., 2025) interpolate frames or construct 3D representations but remain confined to the input's spatiotemporal coverage, essentially performing dynamic novel view synthesis. StarGen (Zhai et al., 2025) scales to long trajectories but assumes static worlds, eliminating temporal dynamics entirely. DFoT (Song et al., 2025) most closely relates to our work, proposing guidance methods that condition on previous frames for scene consistency. However, DFoT also faces fundamental memory constraints when processing extended sequences, limiting its ability to maintain long-range spatial coherence. Our dual spatio-temporal strategy with SLAM-based spatial memory overcomes these limitations by selectively retrieving only the most relevant frames, both temporally and spatially, enabling computationally efficient processing of arbitrary-length videos while maintaining both motion continuity and scene consistency.

## 3 METHODOLOGY

### 3.1 PROBLEM FORMULATION AND MOTIVATION

We address the task of *scene-consistent camera-controllable video generation*: given a dynamic video $\mathbf{V}_{\text{in}} \in \mathbb{R}^{L \times H \times W \times 3}$ of arbitrary length $L$ as context with height $H$ and width $W$, our goal is to generate $T$ subsequent frames $\mathbf{V}_{\text{out}} \in \mathbb{R}^{T \times H \times W \times 3}$ that follow a desired camera trajectory $\mathbf{C} = \{C_t\}_{t=1}^{T}$ while maintaining consistency with the scene captured in the context input:

$$\mathbf{V}_{\text{out}} = \mathcal{F}(\mathbf{V}_{\text{in}}, \mathcal{T}, \mathbf{C}), \tag{1}$$

where $C_t \in \mathbb{SE}(3)$ represents camera extrinsic matrices and $\mathcal{T}$ is a text prompt when a video generator $\mathcal{F}(\cdot)$ is based on pretrained text-to-video priors (Yang et al., 2024; Bahmani et al., 2025a).

**Comparison to existing solutions.** This task fundamentally differs from existing video generation paradigms. Existing camera-controllable generation methods (He et al., 2024; Wang et al., 2024b; Bahmani et al., 2024) synthesize videos following user-specified trajectories but only condition on a single image $\mathbf{I}_{\text{ref}}$ or plain text $\mathcal{T}$ (Fig. 2-(a)):

$$\mathbf{V}_{\text{out}} = \mathcal{F}(\mathbf{I}_{\text{ref}}, \mathcal{T}, \mathbf{C}), \quad \text{or} \quad \mathbf{V}_{\text{out}} = \mathcal{F}(\mathcal{T}, \mathbf{C}), \tag{2}$$

which is insufficient for our task, where the entire underlying 3D scene of the context video should be considered. In contrast, video-to-future-video generation methods such as

`Cosmos-predict-2` (Agarwal et al., 2025) $\mathcal{G}(\cdot)$ employ temporal sliding windows to generate future frames (Fig. 2-(b)):

$$\mathbf{V}_{\text{out}} = \mathcal{G}(\mathbf{V}_{\text{in}}[L - w : L], \mathcal{T}) \tag{3}$$

where $\mathbf{V}_{\text{in}}[L - w : L]$ for $w \ll L$ represents a small overlap window, typically consisting of the last few frames of $\mathbf{V}_{\text{in}}$. Although this design encourages temporal smoothness by providing the last few frames when generating the future video, it often fails to preserve long-term spatial consistency when the camera revisits regions not covered by the small window $w$.

## 3.2 Towards Scene-Consistent Camera-Controllable Video Generation

The key challenge of scene-consistent camera-controllable video generation lies in reconciling two competing requirements: maintaining consistency with potentially distant frames that share spatial proximity (when the camera returns to similar viewpoints), while evolving dynamic content naturally from the recent temporal context. Ideally, conditioning on *all* frames $\mathbf{V}_{\text{in}}$ would ensure optimal global spatial consistency. However, this quickly becomes impractical as the sequence grows, since standard self-attention incurs quadratic time/memory in the sequence length.

**Dual spatio-temporal sliding window strategy.** Instead of increasing the temporal window size $w$ of the existing video-to-future-video generation methods, we introduce a dual sliding window strategy that conditions on frames selected along both *temporal* and *spatial* axes (Fig. 2-(c)). Beyond the standard temporal window that captures recent motion dynamics, we add a spatial window that retrieves frames sharing similar 3D viewpoints, regardless of their temporal distance:

$$\mathbf{V}_{\text{out}} = \mathcal{F}(\tilde{\mathbf{V}}_{\text{in}}, \mathcal{T}, \mathbf{C}), \quad \text{where} \quad \tilde{\mathbf{V}}_{\text{in}} = \{\text{Temporal}(w)\} \cup \{\text{Spatial}(T)\}, \tag{4}$$

where the model $\mathcal{F}$ generates a future sequence $\mathbf{V}_{\text{out}}$ conditioned on Temporal$(w)$, last $w$ frames of the input video $\mathbf{V}_{\text{in}}[L - w : L]$, and Spatial$(T)$, the $T$ retrieved frames from the entire input sequence based on viewpoint similarity to the target viewpoint $\mathbf{C}$. This dual conditioning enables the model to reference distant frames that observe the same spatial regions, maintaining scene consistency without processing all $L$ input frames.

While this dual conditioning is conceptually appealing, naïvely retrieving and providing spatially adjacent frames directly would be problematic for our task. Since we aim to generate future content beyond the input's temporal boundary, directly conditioning on frames from earlier timestamps would incorrectly preserve dynamic elements (e.g., a walking person from frame 50 should not necessarily reappear at that same location when generating frame 200). The spatial conditioning must therefore provide only the persistent scene structure while excluding dynamic content. Rather than retrieving individual frames, we introduce a **3D scene memory** $\mathcal{M}$ that represents exclusively the *static* geometry extracted from all spatially relevant frames.

## 3.3 3D Scene Memory Construction

Our 3D scene memory must efficiently encode spatial relationships across all $L$ frames while extracting only persistent static geometry. To construct the 3D scene memory, we leverage dynamic SLAM frameworks (Li et al., 2024; Zhang et al., 2024) to estimate camera poses and reconstruct 3D structure:

$$(\hat{\mathbf{C}}, \mathbf{P}) = \mathcal{D}_{\text{SLAM}}(V_{\text{in}}), \tag{5}$$

where $\hat{\mathbf{C}} = \{\hat{C}_i\}_{i=1}^{L}$ are the estimated camera poses, $\mathbf{P}$ represents the aggregated 3D point cloud from the $L$ input frames, and $\mathcal{D}_{\text{SLAM}}(\cdot)$ represents the dynamic SLAM framework. This SLAM integration is effective in that it not only estimates the camera parameters of the input frames but also reconstructs the 3D structure of the scene, which can be further utilized to represent the 3D static geometry.

While the camera poses $\hat{\mathbf{C}}$ enable efficient spatial retrieval by comparing viewpoint similarity with the target trajectory $\mathbf{C}$, the aggregated 3D point cloud $\mathbf{P}$ still contains both static and dynamic regions. Thus, we now explain our full pipeline on how to identify dynamic regions and only maintain the persistent static geometry of the input video.

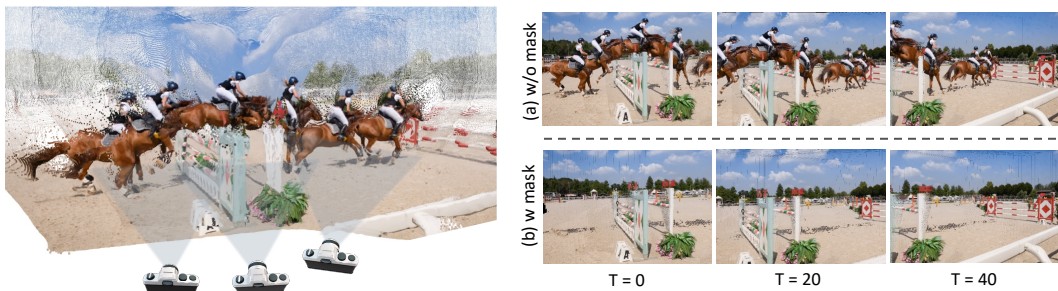

Figure 4: **Illustration of dynamic masking for static scene extraction.** When aggregating 3D points across frames, moving objects create ghosting artifacts if not properly masked. (a) Without masking, dynamic elements (horses and riders) appear frozen at multiple positions, severely degrading the warped views. (b) With our dynamic masking pipeline, these elements are identified and excluded, resulting in clean static-only point clouds that can be reliably warped to new viewpoints.

**Dynamic masking for static scene extraction.** Naïvely aggregating points across frames creates ghosting artifacts where moving objects appear frozen at multiple positions, as shown in Fig. 4-(a). We address this through a comprehensive three-stage masking pipeline that identifies and excludes all dynamic content as depicted in Fig. 5.

We begin with pixel-level motion detection following MonST3R (Zhang et al., 2024). For each frame pair, we compute optical flow using SEA-RAFT (Wang et al., 2024a) ($\text{Flow}_{\text{optical}}$) and compare it against the flow induced by camera motion alone ($\text{Flow}_{\text{warp}}$). Regions where the L1 difference exceeds a specific threshold $\tau$ are marked as potentially dynamic:

$$M_i^{\text{pixel}} = \mathbb{1}\left[\|\text{Flow}_{\text{optical}} - \text{Flow}_{\text{warp}}\|_1 > \tau\right]. \tag{6}$$

However, pixel-level detection captures motion only at specific instants and misses complete object boundaries. We therefore propagate these sparse detections to full objects using SAM2 (Ravi et al., 2024), where we sample points from dynamic pixels in the first frame for prompts. Yet this approach still has limitations: static objects that begin moving in later frames may not be detected if they appear static initially.

Our solution employs backward tracking with CoTracker3 (Karaev et al., 2024) to aggregate motion evidence across the entire sequence. From the sampled points in each frame obtained from our pixel-level motion detection, we track these points from all frames back to $t = 0$, capturing motions of objects that move at any point. These aggregated points are used to prompt the final SAM2 pass, producing complete object-level masks $M_i^{\text{obj}}$ that cleanly separate all dynamic content (Fig. 4-(b)). With the full dynamic mask, we can now obtain the static-only 3D geometry $\mathbf{P}_{\text{static}}$:

$$\mathbf{P}_{\text{static}} = \bigcup_{i=1}^{L} \mathbf{P}_i \odot (1 - M_i^{\text{obj}}). \tag{7}$$

From the constructed static-only 3D geometry $\mathbf{P}_{\text{static}}$ with our proposed dynamic masking strategy, we now obtain the 3D scene memory:

$$\mathcal{M} = (\hat{\mathbf{C}}, \mathbf{P}_{\text{static}}), \tag{8}$$

where we now explain how this 3D scene memory $\mathcal{M}$ can be used for scene-consistent camera-controllable video generation in the following section.

### 3.4 3D Scene Prompting

Having constructed the static-only 3D representation $\mathbf{P}_{\text{static}}$, rather than naïvely retrieving $T$ frames from the input video based on viewpoint similarity, we synthesize static-only spatial frames through the projection of $\mathbf{P}_{\text{static}}$. For each target camera pose $C_t \in \mathbf{C}$, we generate the corresponding spatial frame by projecting the static points from the most spatially relevant input frames:

$$\text{Spatial}(t) = \Pi(K \cdot C_t \cdot \mathbf{P}_{\text{static}}^{(n)}), \tag{9}$$

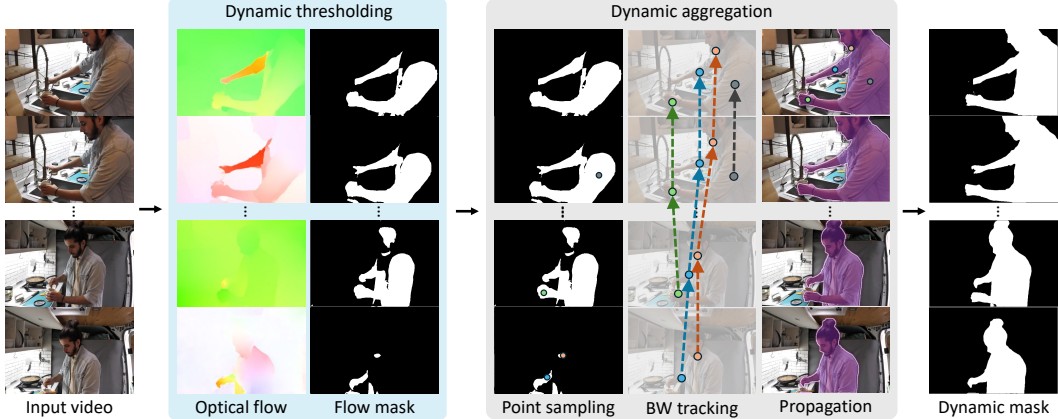

Figure 5: **Dynamic masking strategy.** A three-stage pipeline refines dynamic region detection to produce complete object-level masks: (1) optical-flow differences detect pixel-level motion (Dynamic thresholding); (2) sample points from these regions for all frames and perform backward tracking (BW tracking) with CoTracker3 (Karaev et al., 2024) to aggregate motion evidence across all frames back to $t=0$ (dynamic aggregation), capturing objects that move at any time; (3) propagate aggregated points in the first frame to the entire video using SAM2 (Ravi et al., 2024). The resulting dynamic masks cleanly separate moving elements (people, objects) from the static background, enabling construction of the static-only point cloud $\mathbf{P}_{\text{static}}$.

where $\mathbf{P}_{\text{static}}^{(n)} \subset \mathbf{P}_{\text{static}}$ contains points from the top-$n$ spatially adjacent input frames to $C_t$, $\Pi(\cdot)$ denotes perspective projection, and $K$ is the camera intrinsic matrix. The complete spatial conditioning becomes $\text{Spatial}(T) = \{\text{Spatial}(t)\}_{t=1}^{T} \in \mathbb{R}^{T \times H \times W \times 3}$, where spatial adjacency is calculated by field-of-view overlap.

This projection-based approach ensures only static content appears in conditioning while providing geometrically consistent views aligned to target poses. Notably, the static point cloud aggregates information from multiple viewpoints, potentially filling regions occluded by dynamic objects. These projected views serve as 3D scene prompts that provide explicit guidance about persistent scene structure, enabling precise camera control without additional encoding modules.

The projected views $\text{Spatial}(T)$ serve as what we term *3D scene prompts*—they provide the model with explicit guidance about the persistent scene structure. By conditioning on both $\text{Temporal}(w)$ and $\text{Spatial}(T)$, our framework effectively enables scene-consistent camera-controllable video generation with computational efficiency while preserving the prior for high-quality video synthesis.

# 4 EXPERIMENTS

## 4.1 IMPLEMENTATION DETAILS

**Model architecture.** We build upon CogVideoX-I2V-5B (Yang et al., 2024), extending its single-image conditioning to accept dual spatio-temporal inputs with minimal architectural changes. The key modification is repurposing the existing image conditioning channel to accept concatenated latents from both temporal frames and spatial projections. Specifically, we provide the last $w = 9$ frames from $\mathbf{V}_{\text{in}}$ as temporal conditioning and $T$ projected views from the static point cloud as spatial conditioning. This enables the DiT backbone to remain entirely unchanged, preserving all pretrained video priors. Both conditions are encoded through the frozen 3D VAE and concatenated channel-wise such that $\mathbf{Z}_{\text{cond}} = \mathcal{E}[\text{Concat}(\text{Temporal}(w), \text{Spatial}(T))]$.

**Fine-tuning.** We fully fine-tune the model for a total of 4K iterations with a batch size of 8 using 4 H100 GPUs, which required approximately 48 hours. We used the 16-bit Adam optimizer with a learning rate of $1 \times 10^{-5}$, and adopted the same hyperparameter settings as those used in the training of CogVideoX (Yang et al., 2024). For the temporal sliding window, we provide the last 9 frames of the input video, setting $w = 9$. For the projection of top-$n$ spatially adjacent views, we set $n = 7$.

| Methods | RealEstate10K | | | | DynPose-100K | | | |
|---|---|---|---|---|---|---|---|---|
| | PSNR↑ | SSIM↑ | LPIPS↓ | MEt3R↓ | PSNR↑ | SSIM↑ | LPIPS↓ | MEt3R↓ |
| DFoT (Song et al., 2025) | 18.3044 | 0.5960 | 0.3077 | 0.181164 | 12.1471 | 0.3040 | 0.4172 | 0.183202 |
| **3DScenePrompt (Ours)** | **20.8932** | **0.7171** | **0.2120** | **0.040843** | **13.0468** | **0.3666** | **0.3812** | **0.124189** |

Table 1: **Evaluation of spatial and geometric consistency.** We compare DFoT and our framework on the RealEstate10K (Zhou et al., 2018) and DynPose-100K (Rockwell et al., 2025) datasets. For spatial consistency, we evaluate PSNR, SSIM, and LPIPS on revisited camera trajectories, while for geometric consistency, we report the MEt3R (Asim et al., 2025) metric.

**Experimental settings.** We evaluate our method across four key aspects: camera controllability, video quality, scene consistency, and geometric consistency. Since no prior work directly addresses scene-consistent camera-controllable video generation, we compare against two categories of baselines: (1) camera-controllable methods (CameraCtrl (He et al., 2024), MotionCtrl (Wang et al., 2024b), FloVD (Jin et al., 2025), AC3D (Bahmani et al., 2025a)) for camera control and video quality metrics, and (2) DFoT (Song et al., 2025), which attempts scene-consistent camera-controllable generation, for spatial and geometric consistency metrics.

We primarily evaluate on 1,000 dynamic videos from DynPose-100K (Rockwell et al., 2025). For scene consistency evaluation, we additionally test on 1,000 static videos from RealEstate10K (Zhou et al., 2018), as static scenes provide clearer spatial consistency assessment.

## 4.2 SCENE-CONSISTENT VIDEO GENERATION

**Evaluation Protocol.** As mentioned in Section 3.1, one of the unique and key challenges in scene-consistent camera-controllable video generation is maintaining spatial consistency over extended durations. From a given input video, we evaluate spatial consistency by generating camera trajectories that revisit the viewpoints in the given video. By matching frames in the generated video and the input video that share the same viewpoint, we calculate PSNR, SSIM, and LPIPS. For RealEstate10K, we evaluate the whole image, whereas we only evaluate the static regions by masking out the dynamic regions for DynPose-100K. We also assess geometric consistency using Met3R (Asim et al., 2025), which measures multi-view alignment of generated frames under the recovered camera pose.

**Results.** As shown in Tab. 1, **3DScenePrompt** significantly outperforms DFoT across all metrics for both static and dynamic scenes. Most notably, our MEt3R evaluation error drops 77% (0.041 vs 0.181), demonstrating superior multi-view geometric alignment. While DFoT similarly tackles scene-consistent camera-controllable video generation through history guidance, their approach fails to maintain scene-consistency for long sequences due to memory constraints. In contrast, our dual spatio-temporal conditioning enables long-term scene-consistency without causing significant computational overhead. The qualitative comparisons shown in Fig. 6 also validate the effectiveness of our approach over DFoT.

## 4.3 CAMERA-CONTROLLABLE VIDEO GENERATION

**Evaluation Protocol.** We employ the evaluation protocol of previous methods (He et al., 2024; Zheng et al., 2024; Jin et al., 2025) for the camera controllability. We provide an input image along with associated camera parameters for I2V models (He et al., 2024; Wang et al., 2024b; Jin et al., 2025) and solely provide camera parameters for the T2V model (Bahmani

Table 2: **Camera controllability evaluation.**

| Methods | DynPose-100K | | |
|---|---|---|---|
| | mRotErr (°)↓ | mTransErr↓ | mCamMC↓ |
| MotionCtrl Wang et al. (2024b) | 3.5654 | 7.8231 | 9.7834 |
| CameraCtrl He et al. (2024) | 3.3273 | 9.5989 | 11.2122 |
| FloVD Jin et al. (2025) | 3.4811 | 11.0302 | 12.6202 |
| AC3D Bahmani et al. (2025a) | 3.0675 | 9.7044 | 11.1634 |
| DFoT Song et al. (2025) | 2.3977 | 8.0866 | 9.2330 |
| 3DScenePrompt ($w = 1$) | 2.3898 | 7.7819 | 8.9785 |
| **3DScenePrompt ($w = 9$)** | **2.3772** | **7.4174** | **8.6352** |

et al., 2025a). We conduct experiments using two variants of our framework, leveraging different numbers of frames ($w$) for temporal conditioning baselines, $w = 1$ and $w = 9$. The $w = 1$ uses only the last single frame as temporal conditioning, whereas the $w = 9$ model takes the last nine frames as temporal conditioning. To evaluate how faithfully the generated video follows the camera condition, we estimate camera parameters from the synthesized video using MegaSAM (Li et al., 2024), and compare the estimated camera parameters against the condition camera trajectory **C**.

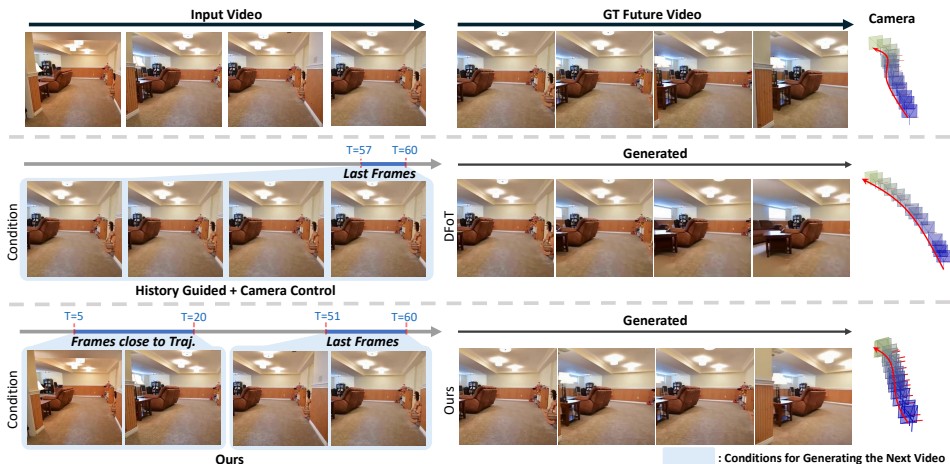

Figure 6: **Visualization of generated videos following trajectories that revisit early frames in the input video.** We visualize and compare frames obtained from DFoT (Song et al., 2025) and Ours. We condition both DFoT and ours to generate a video that follows the camera trajectory shown in the top row (GT Future Video), which revisits a viewpoint in the input (Input Video). The comparison shows that ours shows much more consistent generation, whereas DFoT fails to generate scene-consistent frames mainly due to the limited number of frames it can condition on. Also, the comparison of the camera trajectory shows that our method more faithfully follows the given camera condition.

Table 3: **Evaluation of video generation quality.** We assess the quality of generated videos using FVD and VBench++ scores. For FVD, lower values indicate higher video quality. For VBench++ scores, higher values indicate better performance. All VBench++ scores are normalized.

| Methods | DynPose-100K | | | | | | | | |
|---|---|---|---|---|---|---|---|---|---|
| | FVD | Overall Score | Subject Consist | Bg Consist | Aesthetic Quality | Imaging Quality | Temporal Flicker | Motion Smooth | Dynamic Degree |
| MotionCtrl (Wang et al., 2024b) | 1017.4247 | 0.5625 | 0.5158 | 0.7093 | 0.3157 | 0.3149 | 0.8297 | 0.8432 | 0.7900 |
| CameraCtrl (He et al., 2024) | 737.0506 | 0.6280 | 0.6775 | 0.8238 | 0.3736 | 0.3888 | 0.6837 | 0.6955 | 0.9900 |
| FloVD (Jin et al., 2025) | 171.2697 | 0.7273 | 0.7964 | 0.8457 | 0.4722 | 0.5546 | 0.7842 | 0.8364 | 0.9900 |
| AC3D (Bahmani et al., 2025a) | 281.2140 | 0.7428 | 0.8360 | 0.8674 | 0.4766 | 0.5381 | 0.8020 | 0.8673 | 1.0000 |
| **3DScenePrompt (Ours)** | **127.4758** | **0.7747** | **0.8669** | **0.8727** | **0.4990** | **0.5964** | **0.8551** | **0.9260** | 1.0000 |

The comparison between the estimated and input camera parameters is quantified using three metrics: mean rotation error (mRotErr), mean translation error (mTransErr), and mean error in the camera extrinsic matrices (mCamMC). For the generated video, we also assess video synthesis performance using the Fréchet Video Distance (FVD) (Skorokhodov et al., 2022) and seven metrics from VBench++ (Huang et al., 2024): subject consistency, background consistency, aesthetic quality, imaging quality, temporal flickering, motion smoothness, and dynamic degree.

**Results.** We first evaluate camera controllability and compare our method with competitive baselines. As shown in Tab. 2, our approach consistently outperforms existing methods, indicating **3DScenePrompt** is capable of generating videos with precise camera control. We also note that the effect of using different temporal conditioning window sizes is minimal for the camera-controllability performance, suggesting that better camera-controllability is achieved from our spatial prompts rather than the increased temporal context. We then assess the overall video quality (Tab. 3) and provide qualitative comparisons (Fig. 7). As observed in Tab. 3, our method achieves the best generation quality across all metrics for dynamic video generation, which is further supported by the visual results in Fig. 7.

## 4.4 ABLATION STUDIES

We analyze two critical components of our framework: the dynamic masking strategy that separates static and dynamic elements, and the number of spatially adjacent frames $n$ retrieved for spatial conditioning. Tab. 4 demonstrates the impact of varying $n$ and the necessity of dynamic masking.

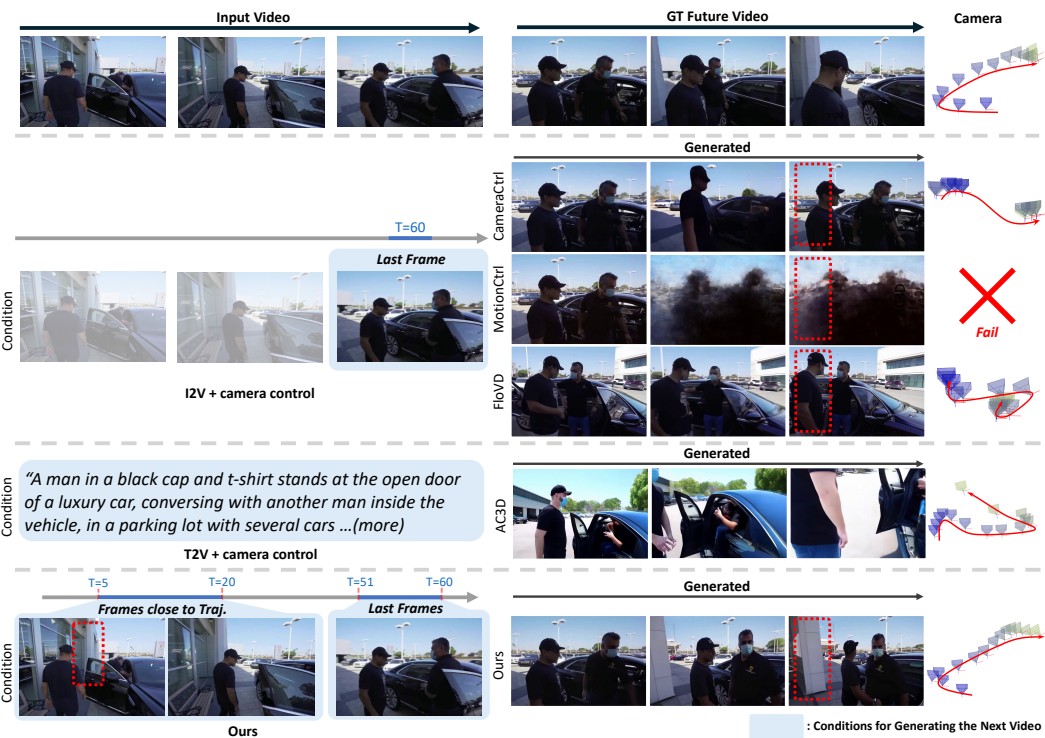

Figure 7: **Visualization of scene-consistent camera-controllable video generation.** Comparison of different methods (Wang et al., 2024b; He et al., 2024; Jin et al., 2025; Bahmani et al., 2025b) for generating videos from the same input (shown in Input Video) that follow the camera trajectory shown in the rightmost column (Camera). Our method best preserves scene consistency with the input video. Note the red-boxed regions: while the input video shows a white wall, competing methods either lose scene detail or fail to maintain the original scene structure. In contrast, our approach accurately remembers the white wall and maintains consistent scene elements throughout generation. In addition, when compared with the GT Future Video, ours best follows the camera condition, effectively verifying the strength of our framework for scene-consistent camera-controllable video generation.

Without dynamic masking (4th row), the model suffers significantly across all, showing a large drop of PSNR of approximately 0.8dB and also an increase of MEt3R error.

This degradation occurs because unmasksed dynamic objects create ghosting artifacts when warped to new viewpoints, corrupting the spatial conditioning. Regarding the number of spatially adjacent frames, we find that performance stabilizes around $n = 7$, with minimal improvements beyond this point, suggesting that 7 frames provide sufficient spatial context while maintaining computational efficiency.

Table 4: **Ablation study on varying $n$.**

| Methods | Dynamic mask $\mathcal{M}$ | DynPose-100K | | | |
|---|---|---|---|---|---|
| | | PSNR↑ | SSIM↑ | LPIPS↓ | MEt3R↓ |
| **Ours** ($n = 1$) | ✓ | 13.0207 | 0.3732 | 0.3771 | 0.124773 |
| **Ours** ($n = 4$) | ✓ | 13.0382 | 0.3733 | 0.3758 | 0.124893 |
| **Ours** ($n = L$) | ✓ | 13.0206 | 0.3631 | 0.3810 | 0.123507 |
| **Ours** ($n = 7$) | ✗ | 12.2304 | 0.3063 | 0.3821 | 0.134885 |
| **Ours** ($n = 7$) | ✓ | 13.0468 | 0.3666 | 0.3812 | 0.124189 |

## 5 CONCLUSION

In this work, we introduced **3DScenePrompt**, a framework for scene-consistent camera-controllable video generation. By combining dual spatio-temporal conditioning with a static-only 3D scene memory constructed through dynamic SLAM and our dynamic masking strategy, we enable generating continuations from arbitrary-length videos while preserving scene geometry and allowing natural motion evolution. Extensive experiments demonstrate superior performance in camera controllability, scene consistency, and generation quality compared to existing methods. Our approach opens new possibilities for long-form video synthesis applications where maintaining both spatial consistency and precise camera control is essential.

ACKNOWLEDGMENTS

This research was supported by Institute of Information & communications Technology Planning & Evaluation (IITP) grant funded by the Korea government (MSIT) (RS-2019-II190075, RS-2024-00509279, RS-2025-II212068, RS-2023-00227592, RS-2025-02214479, RS-2024-00457882, RS-2025-25441838, RS-2025-25441838, RS-2025-02214479, RS-2025-02217259) and the Culture, Sports, and Tourism R&D Program through the Korea Creative Content Agency grant funded by the Ministry of Culture, Sports and Tourism (RS-2024-00345025, RS-2024-00333068, RS-2023-00222280, RS-2023-00266509), and National Research Foundation of Korea (RS-2024-00346597), and the ETH AI Center through an ETH AI Center postdoctoral fellowship to Sunghwan Hong.

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

# APPENDIX

## NOTATION SUMMARY

| Symbol | Meaning |
|---|---|
| $V_{\text{in}} \in \mathbb{R}^{L \times H \times W \times 3}$ | input arbitrary-length video with $L$ frames. |
| $V_{\text{out}} \in \mathbb{R}^{T \times H \times W \times 3}$ | generated future video with $T$ frames. |
| $\mathcal{T}$ | text prompt for video generation for video models based on text-to-video (T2V) generation. |
| $\mathbf{C} = \{C_t\}_{t=1}^{T}$ | desired camera trajectory the generated video $V_{\text{out}}$ should follow. |
| $C_t$ | camera extrinsics parameter where $C_t \in \mathbb{SE}(3)$. |
| $K$ | camera intrinsics parameter. |
| $\mathcal{F}(\cdot)$ | camera-controllable video generation framework. |
| $\mathcal{G}(\cdot)$ | video-to-future-video generation framework. |
| $\mathbf{I}_{\text{ref}}$ | image condition for image-to-video (I2V) generation. |
| $V[x:y]$ | indexing operation; samples frames between frame x and (y-1). |
| $\text{Temporal}(w)$ | temporally adjacent $w$ frames for condition. |
| $\text{Spatial}(T)$ | spatially adjacent $T$ frames for condition. |
| $\tilde{V}_{\text{in}}$ | conditioning frames for our framework, includes both $\text{Temporal}(w)$ and $\text{Spatial}(T)$. |
| $\mathcal{D}(\cdot)$ | dynamic SLAM frameworks. |
| $\Pi(\cdot)$ | perspective projection operator. |
| $\mathbf{P}$ | aggregated 3D point clouds. |
| $\mathbf{P}_{\text{static}}$ | aggregated 3D point clouds only from static regions. |
| $\mathcal{M}$ | 3D scene memory composed of camera extrinsics $C_t$ and static point clouds $\mathbf{P}_{\text{static}}$. |
| $M_i^{\text{obj}}$ | object-level masks representing dynamic regions of frame $i$. |
| $\mathcal{E}(\cdot)$ | 3D VAE. |

## A  ADDITIONAL EXPERIMENTAL RESULTS

**More Qualitative Results.**  We provide additional qualitative results of our method to further demonstrate its ability to generate high-quality outputs, as shown in 9, 10, and 11.

**Long-Video Generation Results.**  Although our primary interest is building a framework capable of generating a spatially consistent next-video chunk given an arbitrary video as context, by iteratively applying our method, one of the potential applications of our framework is generating videos of arbitrary length.

To generate longer videos, we extend our method using iterative autoregressive generation, and conduct quantitative evaluations on long-video generation using the DAVIS dataset (Perazzi et al., 2016), where we evaluate video generation quality (PSNR, SSIM, LPIPS) and camera-controllability (RMSE, MSE, ATE), and compare with previous I2V baselines as reported in Tab. 5. However, as there are no publicly available video-to-future-video camera-controllable generation methods for our baselines, for a fairer comparison, we adopt three different strategies to generate long videos with our baseline methods. Specifically, we adopt **1)** iterative autoregressive generation, **2)** latent interpolation with autoregressive generation, and  **3)** applying training-free long video generation techniques.

For iterative autoregressive generation, we simply provide the last frame of the previously generated video as the input condition when generating the next video. For latent interpolation, we increase the number of input latents through interpolation, which can directly increase the number of generated frames at inference. After applying latent interpolation, we similarly adopt autoregressive generation to generate videos with the desired number of frames. By generating more frames at once, this reduces the number of iterations of autoregressive generation, which can increase overall video quality by reducing error accumulation. Finally, we also adopt FreeLong (Lu et al., 2024), a training-free approach for long video generation, which introduces frequency blending of latents specifically in Stable Video Diffusion (Blattmann et al., 2023). As we compare with the FloVD (Jin et al., 2025) model fine-tuned from CogVideoX (Yang et al., 2024), which is a transformer-based

Table 5: **Quantitative evaluation of spatially consistent long video generation on DAVIS (Perazzi et al., 2016) dataset.**

| Methods | DAVIS | | | | | |
|---|---|---|---|---|---|---|
| | PSNR↑ | SSIM↑ | LPIPS↓ | RMSE↓ | MSE↓ | ATE↓ |
| CameraCtrl (He et al., 2024) | 8.64 | 0.19 | 0.66 | 95.45 | 9332.08 | **0.1709** |
| + latent interpolation | 14.12 | 0.49 | 0.45 | 53.50 | 3214.71 | 0.2225 |
| + FreeLong (Lu et al., 2024) | 13.46 | 0.44 | 0.48 | 55.84 | 3314.31 | 0.2147 |
| FloVD (Jin et al., 2025) | 10.77 | 0.42 | 0.55 | 76.08 | 6131.95 | 0.2242 |
| Ours | **17.28** | **0.60** | **0.35** | **37.28** | **1583.77** | 0.1794 |

Table 6: **Dynamic mask ablation study.**

| Methods | DynPose-100K | | | | | | |
|---|---|---|---|---|---|---|---|
| | PSNR↑ | SSIM↑ | LPIPS↓ | MEt3R↓ | mRotErr (°)↓ | mTransErr↓ | mCamMC↓ |
| (a) w/o dynamic mask | 11.9963 | 0.2898 | 0.3748 | **0.104002** | 3.4142 | 8.5920 | 10.5049 |
| (b) (a) + L1 difference mask | 12.1083 | 0.3248 | 0.3690 | 0.155589 | 2.9390 | 7.7819 | 9.1524 |
| (c) (b) + SAM2 propagation | 13.5957 | 0.4036 | 0.3548 | 0.157610 | 2.7188 | 7.5402 | 8.9696 |
| (d) (c) + point BW tracking | **13.7311** | **0.4112** | **0.3454** | 0.122859 | **2.6103** | **7.4858** | **8.9181** |

model, we apply FreeLong only for CameraCtrl (He et al., 2024). Due to latent interpolation significantly increasing the computation in CogVideoX, we also apply the latent interpolation technique for CameraCtrl only. The qualitative results of our generated long videos can be found in the `supplementary videos` or in Fig. 12.

As shown in the results, our method shows its potential for long video generation, significantly outperforming previous baselines. However, as is common in auto-regressive generation, extending our framework to extremely long durations introduces the challenge of temporal error accumulation. While our method provides the critical ***spatial consistency*** required for long videos, solving long-term drift (error accumulation) typically requires dedicated refinement modules and techniques. We believe that further extending our framework to mitigate the aforementioned issues is a very important and interesting future direction to explore.

**Ablation Study on Dynamic Masking Pipeline.** We ablate each stage of our dynamic masking pipeline presented in Sec. 3.3. Stage (a) performs training without any dynamic mask, causing moving objects to interfere with static-scene learning. Stage (b) introduces a pixel-level L1 flow-difference mask, as described in Eq. 6, which detects motion per frame but often fails to fully capture dynamic objects due to noisy or fragmented flow estimation. Stage (c) improves object-level consistency by using SAM2 propagation from points sampled in the first frame; however, this stage cannot detect new or displaced dynamic objects appearing in later frames. Stage (d) further adds point tracking across the sequence, enabling backward propagation into the first frame before SAM2 segmentation, which allows the mask to capture all dynamic objects throughout the entire clip. The quantitative comparison for each variant is summarized in Tab. 6, where we evaluate the video generation quality (PSNR, SSIM, LPIPS), scene-consistency (Met3r), and camera-controllability (mRotErr, mTransErr, mCamMC) in a subset of the DynPose-100K dataset. The results effectively verify the need for each stage of our proposed dynamic mask generation pipeline.

**Ablation study on the number of temporal condition images.** As CogVideoX generates 49 frames simultaneously, the temporal window size $w$ can be selected within the range $1 < w < 49$, meaning $(49 - w)$ frames are newly generated. In our framework, setting the window size to $w = 9$ (providing the last few frames) is one of the key contributions and design choices to ensure sufficient temporal context to maintain motion continuity and coherency with the input video, while still generating a sufficient number of frames (40). To better validate this choice, we conduct experiments with varying window sizes ($w = 1$ and $w = 5$). As summarized in Tab. 7, the results demonstrate that while the window size has a negligible effect on camera controllability metrics (mRotErr, mTransErr, mCamMC), providing a larger context window ($w = 9$) yields superior motion smoothness and temporal coherence, as evidenced by improved VBench metrics (Temporal Flicker and Motion Smoothness), which is aligned with our intentions.

Table 7: **Ablation study on the size of temporal window** $w$. We conduct an ablation study on camera controllability with respect to the number of conditioned images $w$, and evaluate motion coherence by measuring VBench (Huang et al., 2024)'s Temporal Flicker and Motion Smooth metrics. VBench metrics are normalized.

| Methods | DynPose-100K | | | | |
|---|---|---|---|---|---|
| | mRotErr↓ | mTransErr↓ | mCamMC↓ | Temporal Flicker↑ | Motion Smooth↑ |
| Ours ($w = 1$) | 2.3898 | 7.7819 | 8.9785 | 0.8379 | 0.9253 |
| Ours ($w = 5$) | 2.3837 | 7.5512 | **8.6233** | 0.8508 | 0.9262 |
| Ours ($w = 9$) | **2.3772** | **7.4174** | 8.6352 | **0.8561** | **0.9335** |

Table 8: **Ablation study on the number of projected images** $n$. We conduct an ablation study on scene consistency and camera controllability with respect to the number of spatially adjacent frames $n$ retrieved for spatial conditioning on the DynPose-100K (Rockwell et al., 2025) dataset.

| Methods | DynPose-100K | | | | | |
|---|---|---|---|---|---|---|
| | PSNR↑ | SSIM↑ | LPIPS↓ | mRotErr↓ | mTransErr↓ | mCamMC↓ |
| Ours ($n = 0$) | 11.9555 | 0.3370 | 0.4512 | 3.4142 | 8.5920 | 10.5049 |
| Ours ($n = 4$) | 13.0382 | **0.3733** | **0.3758** | **2.3739** | 7.4278 | 8.6488 |
| Ours ($n = 7$) | **13.0468** | 0.3666 | 0.3812 | 2.3772 | **7.4174** | **8.6352** |

**Ablation study on the number of projected images.** We further evaluate the impact of using different numbers of projected views as spatial conditions. To do this, we compare our method against a baseline setting ($n = 0$), where $n$ denotes the number of projected frames. In this baseline, the model is conditioned solely on the temporal frames without any projected spatial views. The results, presented in Tab. 8, show that without the incorporation of our spatial prompts, the baseline framework fails to maintain scene consistency (resulting in significantly lower PSNR and SSIM scores) and struggles to adhere to the target camera trajectory. In contrast, the variants utilizing projected images ($n = 4$ and $n = 7$) significantly outperform the $n = 0$ baseline across all metrics. This effectively verifies the critical importance of our spatial prompts for ensuring both high-fidelity scene consistency and precise camera control.

**DepthAnything v3 for 3D memory.** One of the advantages of our framework is that we do not have any special architecture designs tailored to MegaSAM and can always replace MegaSAM with any geometry or correspondence models Hong et al. (2022a; 2024c;a); Wang et al. (2025); Cho et al. (2021; 2022), typically a more robust and powerful model to resolve the current limitations. Here, we show an experiment where we replace MegaSAM with the recently released DepthAnything v3 (Lin et al., 2025a) for inference. The results, summarized in Tab. 9, show improvements across metrics, together with reduced inference time. This confirms that our method directly benefits from stronger priors and highlights a key advantage of our design: the components can be swapped at inference time without changing the overall pipeline, allowing users to flexibly choose different pretrained models.

**Inference Time.** Tab. 10 presents the end-to-end inference latency of our method in comparison to existing camera-controllable video diffusion models. Approaches such as CameraCtrl (He et al., 2024), MotionCtrl (Wang et al., 2024b), and FloVD (Jin et al., 2025) rely exclusively on diffusion-based synthesis, where inference time is primarily determined by the denoising process. In contrast, our pipeline incorporates additional stages for SLAM-based 3D reconstruction and dynamic masking. With MegaSAM (Li et al., 2024) employed for dynamic SLAM, the preprocessing stage requires approximately 4 minutes, resulting in a total inference time of over 9 minutes. However, replacing MegaSAM with a more lightweight and advanced method, such as DepthAnything v3 (Lin et al., 2025a), significantly reduces the SLAM processing time to roughly 10 seconds. This simple replacement reduces the total latency to approximately 5 minutes, where the overall runtime is dominated by the diffusion-based video generation itself. Consequently, our framework achieves inference speeds comparable to CameraCtrl and MotionCtrl, which leverage a more lightweight video generation backbone, demonstrating that the proposed method introduces negligible computational overhead while enabling spatially consistent and camera-controllable generation.

Table 9: **Using various methods for generating 3D memory.**

| Methods | DynPose-100K | | |
|---|---|---|---|
| | PSNR↑ | SSIM↑ | LPIPS↓ |
| Ours w/ MegaSAM (Li et al., 2024) | 13.0468 | 0.3666 | 0.3812 |
| Ours w/ DepthAnything v3 (Lin et al., 2025a) | **13.4534** | **0.3980** | **0.3637** |

Table 10: **Inference time comparison of different methods.**

| Method | SLAM-Processing | Dynamic Masking & Depth Warping | Video Generation | Inference Time |
|---|---|---|---|---|
| CameraCtrl (He et al., 2024) | – | – | 1 min 38 sec | 1 min 38 sec |
| MotionCtrl (Wang et al., 2024b) | – | – | 2 min 9.5 sec | 2 min 9.5 sec |
| FloVD (Jin et al., 2025) | – | – | 8 min 5.32 sec | 8 min 5.32 sec |
| Ours – MegaSAM (Li et al., 2024) | 4 min 18.813 sec | 58.88 sec | 4 min 3.62 sec | 9 min 21.31 sec |
| Ours – DepthAnything v3 (Lin et al., 2025a) | 10.031 sec | 58.88 sec | 4 min 3.62 sec | 5 min 12.53 sec |

# B   TRAINING DATASET CURATION PIPELINE.

Our training data is curated from two primary sources: RealEstate10K (Zhou et al., 2018), which consists of static indoor scenes, and OpenVid-1M (Nan et al., 2024), which features diverse dynamic content. For static videos from RealEstate10K, we directly extract 3D scene geometry without applying dynamic masking, as these scenes contain negligible motion. In contrast, for dynamic videos from OpenVid-1M, we conduct extensive data filtering and preprocessing: we remove game-like or low-resolution videos, exclude samples with excessive camera motion, and apply the dynamic masking pipeline described in Section 3.3 to separate static and dynamic regions. We further select long video sequences with lengths of at least $L \geq 100$ frames to ensure sufficient input video length. To construct the 3D scene memory, we employ VGGT (Wang et al., 2025) for static scenes and MegaSAM (Li et al., 2024) for dynamic scenes. After filtering and processing, our final dataset comprises approximately 30K and 20K high-quality long videos from RealEstate10K and OpenVid-1M, respectively.

# C   DISCUSSION AND COMPARISON WITH RELATED WORKS

In this section, we discuss the differences between our proposed framework and recent relevant works, focusing on 3D memory integration, camera-controllable generation, and dynamic object handling.

**Explicit 3D Memory and World Models**   Several recent works have explored utilizing 3D memory for video generation. Persistent embodied world model (Zhou et al., 2025) introduces 3D memory but relies on implicit memory representations Hong et al. (2022b) and targets *static* scenes. In contrast, our work leverages explicit 3D memory to achieve scene-consistent generation specifically for *real-world dynamic* videos. Similarly, WorldMem (Xiao et al., 2025) builds upon the Oasis (Decart et al., 2024) architecture and is trained primarily on the Minecraft dataset. It addresses static scenes (RealEstate10K) or synthetic domains and requires key frames as conditions by increasing the sequence size. Our approach differs by leveraging the learned priors of DiTs to generate scene-consistent, camera-controllable videos in complex real-world dynamic scenarios. Furthermore, by projecting constructed static-only 3D point clouds to the target trajectory, we condition on long videos efficiently without increasing computational costs.

We also differentiate our work with SPMem (Wu et al., 2025), a concurrent work conceptually similar to ours. However, our method distinguishes itself in two key aspects. First, regarding dynamic object handling, while SPMem computes static components via naïve TSDF fusion, we introduce a concise and accurate dynamic mask generation pipeline to effectively remove dynamic regions. Second, in terms of efficiency, SPMem employs an additional ControlNet-style Diffusion-as-Shader (Gu et al., 2025) architecture, necessitating architecture search, such as the number of sufficient layers for the ControlNet-style block for different diffusion backbones, and the introduction of the additional model leads to extra computational resources for training. Conversely, we achieve scene-consistent

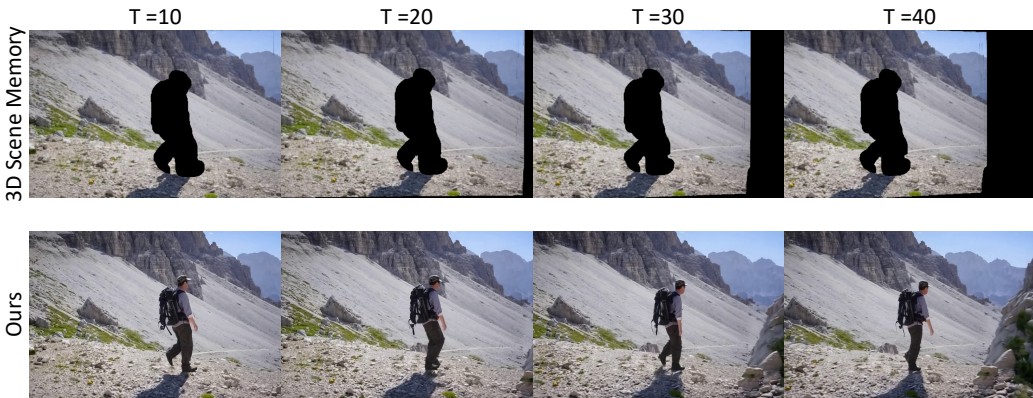

Figure 8: **Visualization of the 3D scene memory and the generated results.**

generation without any architectural changes by injecting spatial and temporal conditioning into zero-padding slots, enabling efficient training and inference. While a direct comparison in performance and efficiency would better highlight these differences, it is currently not possible due to the unavailability of public code.

**Camera-Controllable Video Generation** Works such as CameraCtrl2 (He et al., 2025a) and APT2 (Lin et al., 2025b) explore camera conditioning using Plücker embeddings and autoregressive generation. While effective for short sequences, these methods typically model consistent video generation by taking only previous images as conditions without explicitly modeling 3D scene structure or considering spatial adjacency. Consequently, they fail to incorporate all previous frames due to computational constraints.

In contrast, our approach constructs an explicit 3D static memory using point clouds Yue et al. (2025), enabling precise pixel-wise spatial alignment across views. Instead of relying on autoregressive propagation or feature-level Plücker conditioning, our 3D memory is directly integrated into the diffusion transformer via 3D-warped point clouds, providing scene-consistent constraints that do not accumulate drift over time. Additionally, our design explicitly handles dynamic objects by separating static and dynamic components, a challenge not addressed by these prior methods.

**Dynamic Object Removal and Inpainting** Regarding the removal of dynamic objects, our spatial conditioning shares surface-level similarities with video inpainting techniques such as FGVC (Gao et al., 2020). However, a fundamental difference lies in the role of the condition, as shown in Fig. 8. In standard inpainting, the model operates under hard constraints to fill masked regions. In our framework, spatial prompts serve as a *soft spatial guide*. This allows the diffusion model to fill holes and refine details while respecting the geometric layout where reliable projections exist. Crucially, this flexibility ensures the model can generate dynamic details on top of the condition, blending geometric consistency with realistic temporal dynamics, rather than merely filling static holes.

# D LIMITATION

Our framework, **3DScenePrompt**, enables scene-consistent, camera-controllable video generation and demonstrates improved performance over existing methods. Nonetheless, it has a few limitations. To obtain point clouds, we rely on MegaSAM (Li et al., 2024) to process the input videos, which introduces additional computational overhead and increases inference time. This overhead could be reduced by adopting a more advanced module, *e.g.*, DepthAnything3 (Lin et al., 2025a). In addition, the quality of the dynamic masks directly affects the fidelity of the generated videos. Thanks to our general and modular design, replacing the current component with a more advanced dynamic masking strategy that benefits from stronger models could further improve overall performance.

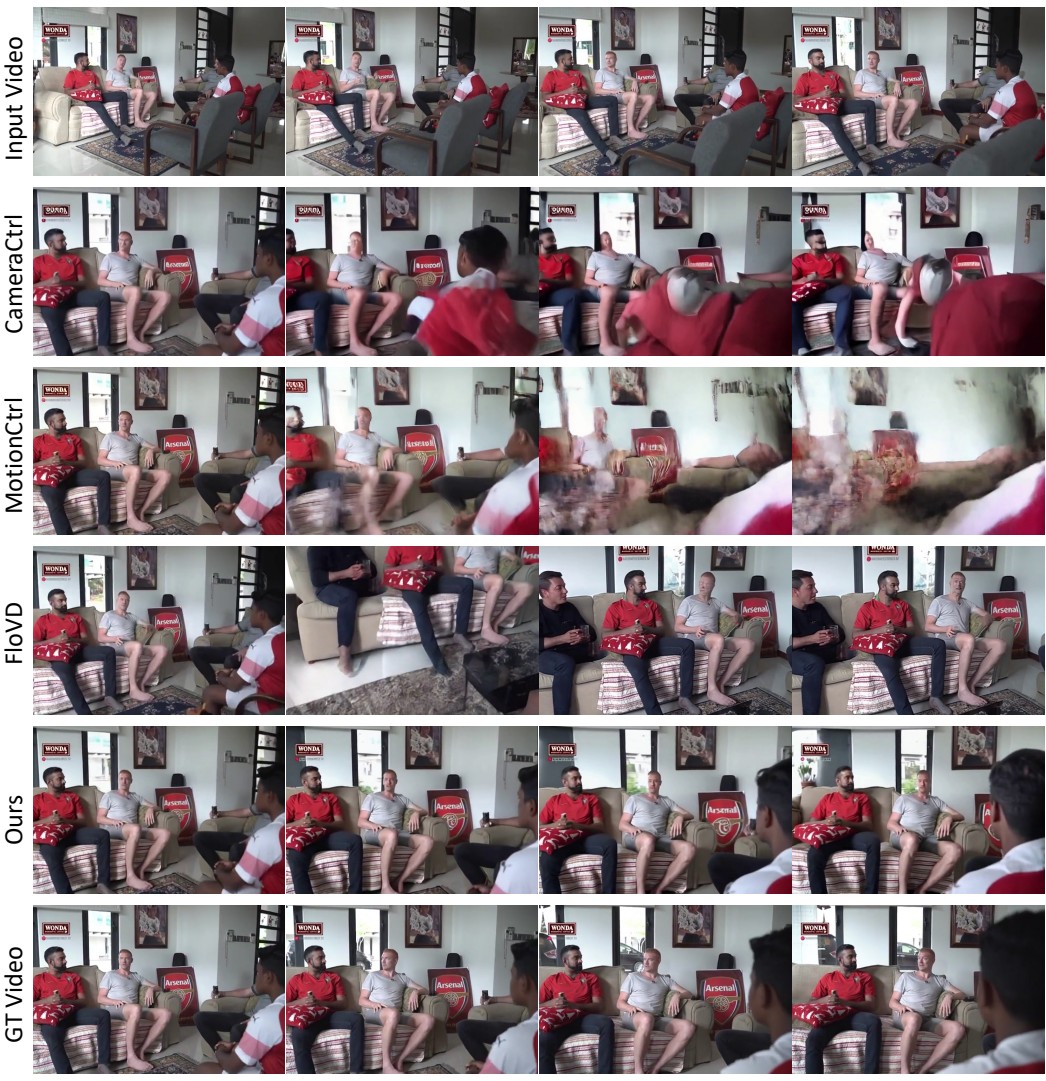

Figure 9: **More qualitative results.**

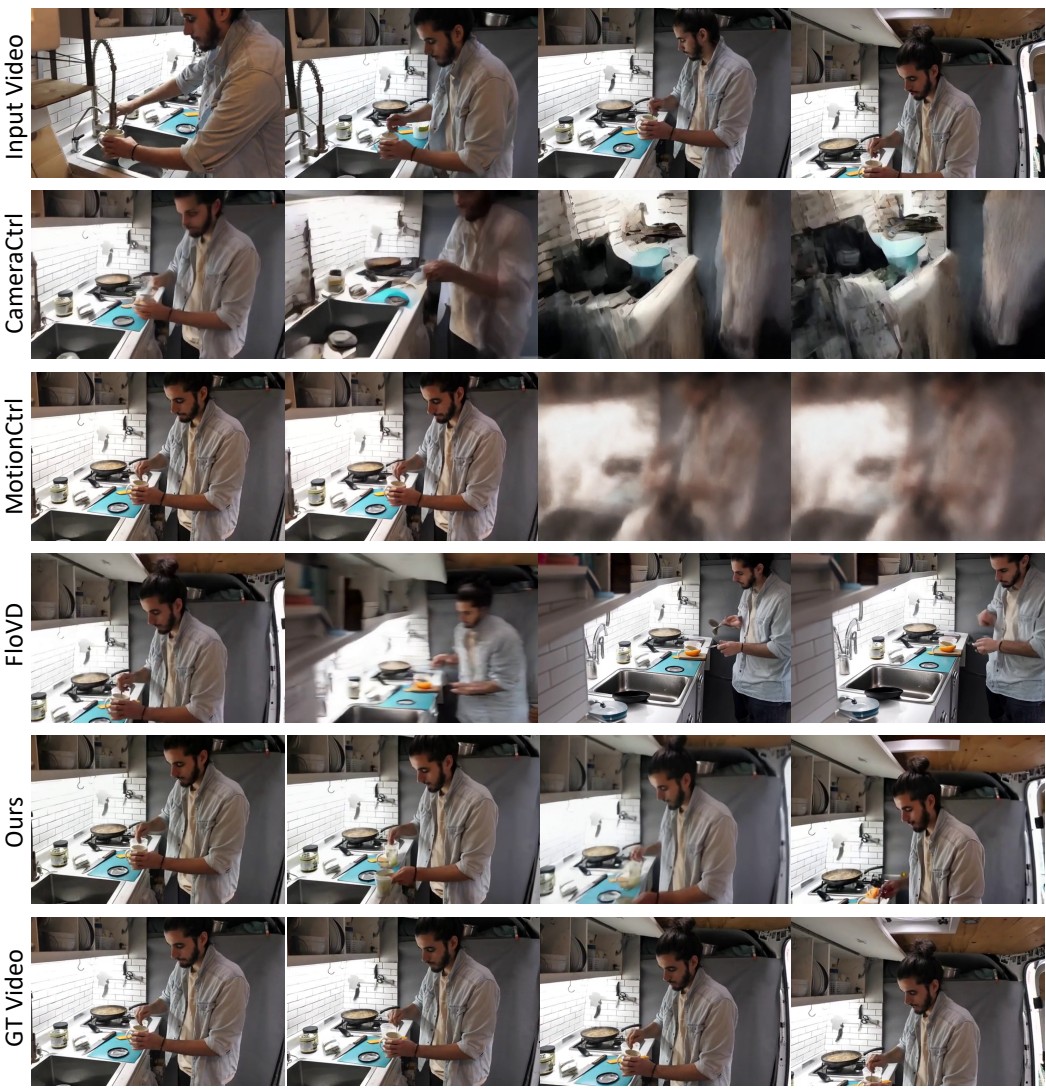

Figure 10: **More qualitative results.**

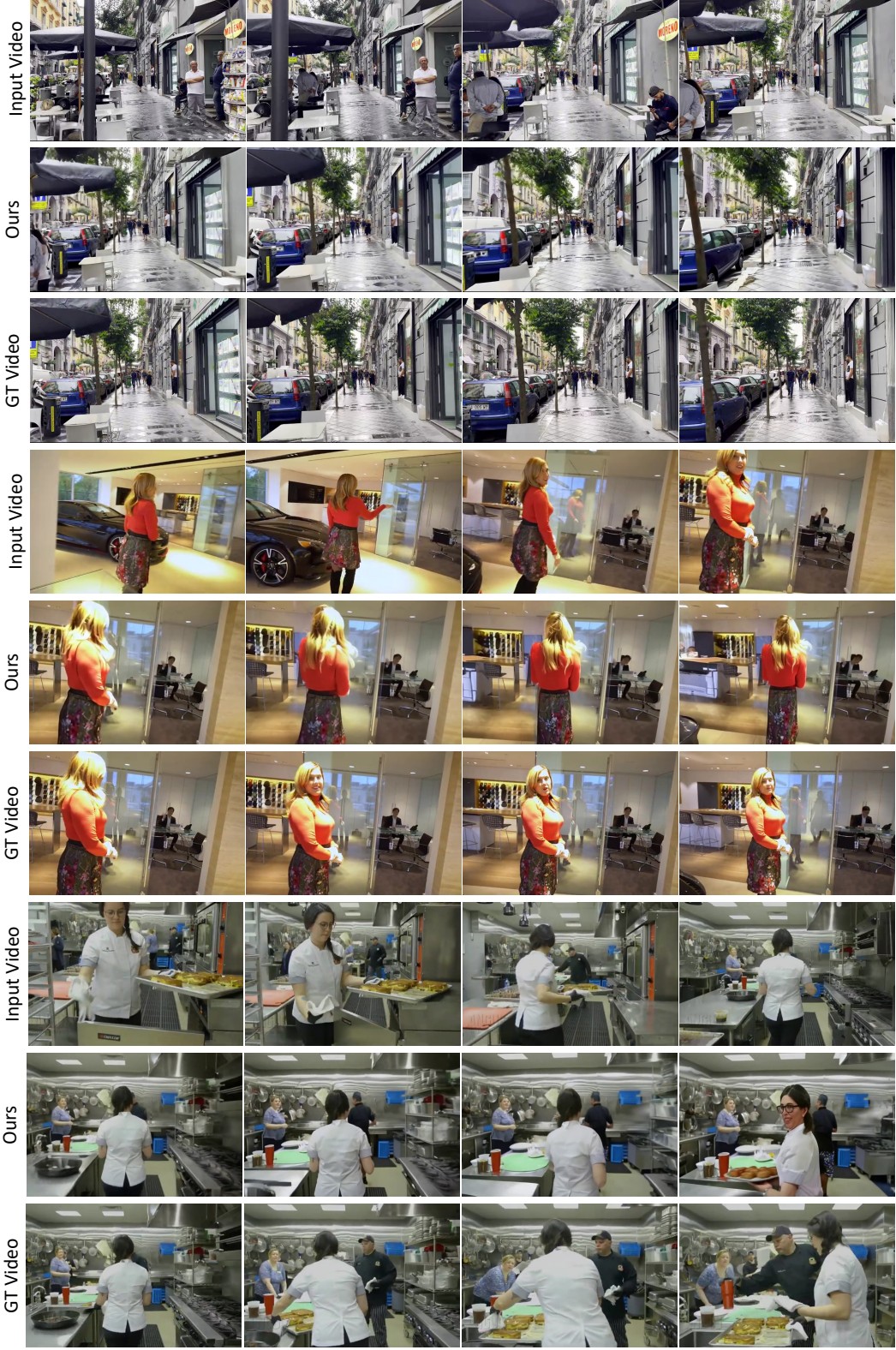

Figure 11: **More qualitative results.**

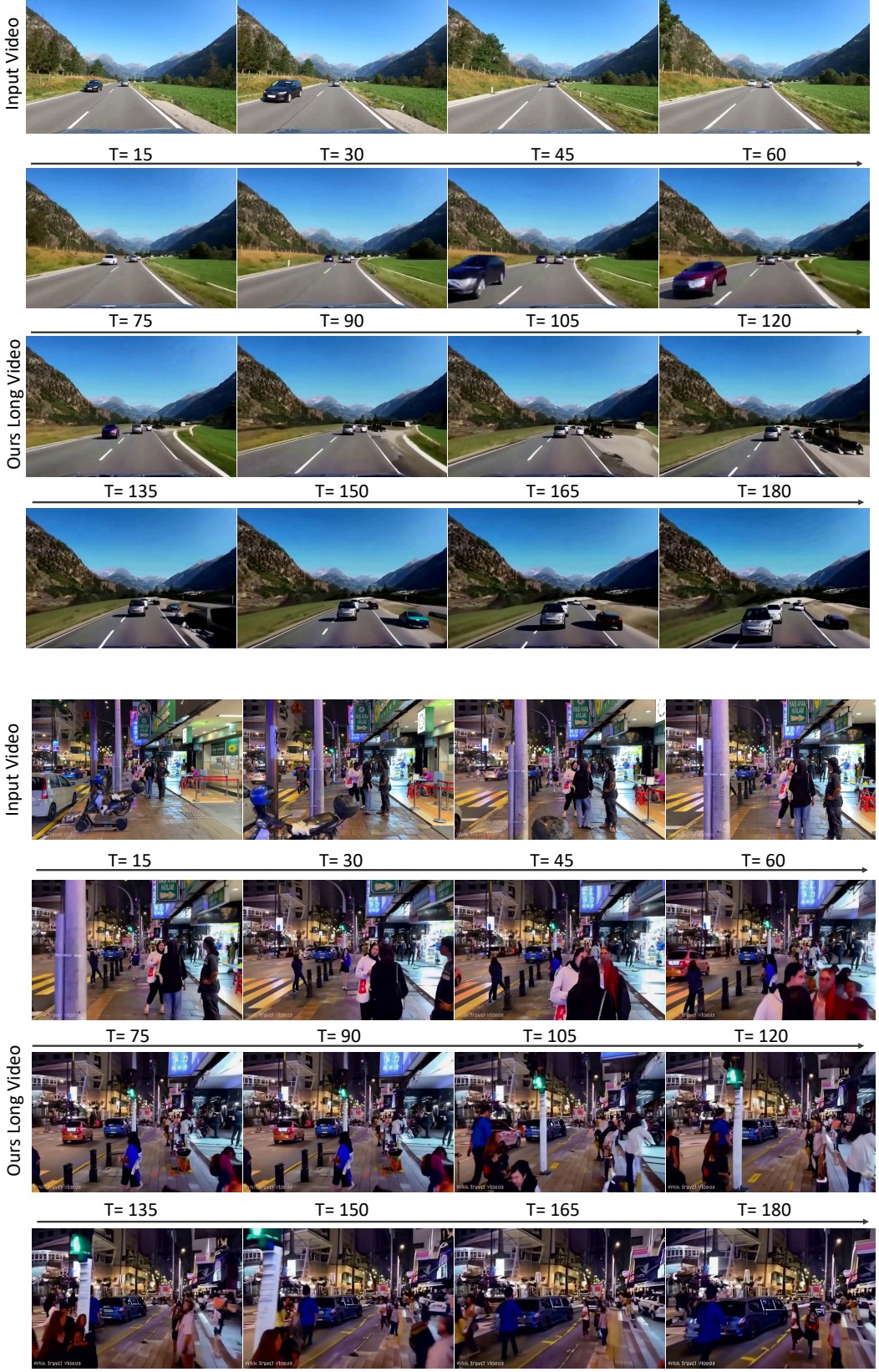

Figure 12: **Long-video generation results.**

# E  REPRODUCIBILITY STATEMENT

As mentioned in Section 3.1, our model builds upon the open-sourced CogVideoX-I2V-5B (Yang et al., 2024) model, where each of the processes for dynamic masking is also detailedly explained. We will also make all the codes publicly available.

# F  USE OF LARGE LANGUAGE MODELS

In accordance with the ICLR 2026 submission policy, we disclose that we used Large Language Models to assist in grammar correction for the writing in this manuscript.

