# OpenReview forum: "3D Scene Prompting for Scene-Consistent Camera-Controllable Video Generation"
_ICLR.cc/2026/Conference — ICLR 2026 Poster_

### Official Review · Reviewer_xUs1 · 2025-10-26

**Soundness:** 3
**Presentation:** 3
**Contribution:** 4
**Rating:** 6
**Confidence:** 4

**Summary:**

The method conditions a video diffusion backbone on both the last few temporal frames for motion continuity and projected views from a static‑only 3D scene memory for spatial consistency, where the memory is created by dynamic SLAM plus a masking pipeline that removes moving objects using optical‑vs‑warped flow differencing, backward point tracking, and SAM2 propagation to yield clean static point clouds. Static‑only point clouds are projected from spatially adjacent viewpoints to the target camera poses and concatenated with temporal latents through a frozen 3D VAE channel, enabling camera‑controllable generation without modifying the DiT backbone.

**Strengths:**

1. Proposed new problem statement which is relevant to Video Generation which the previous camera controllable methods which conditions only on the last few frames
2. Dual spatio‑temporal conditioning addresses both motion continuity and scene consistency, overcoming the short‑window limitation of prior future‑video models.
3. A static‑only 3D scene memory cleanly separates persistent geometry from dynamics, avoiding the leakage of outdated moving objects into future frames.
4. Three‑stage dynamic masking (optical‑vs‑warped flow differencing, backward point tracking, SAM2 propagation) yields clean static point clouds without ghosting.

**Weaknesses:**

1. In the first video in the supplementary video, I see that the spatial structure is not preserved. In the initial video, there were supposed to be 2 chairs side-by-side but I don’t see them in the generated video even though the 3D scene memory captures it. Instead a white pillor / cupboard seems to appear which is wrong. Why is it so? Is there any error in camera pose estimation that when back projected on image space, they are not projected accurately and may lead to inconsistency?
2. I feel like the comparisons are not fair. The video comparisons are all against methods which only do camera control video generation with only the last frame as prior. But your method takes in the few initial videos frames along with the last few frames for video generation and so automatically your method would be more consistent since it is taking more frames as input. This is an unfair comparison in my opinion. The good qualitative results maybe just because you are conditioning your model with more priors and hence more consistent results or is it because of your proposed new architectural changes that enabled better consistent results. I feel the paper didn’t differentiate these two effectively. I agree that it is a novel problem statement and hence no baselines exist but the authors could have trained CameraCtrl or any of the best model to take in more inputs and trained the model with very minimal changes to effectively clarify if the difference in clarity is from more data inputs or actual model architectural changes.
3. There are no limitations mentioned in the main paper or the appendix. I request the authors to propose the limitations of the work and the future work. It is important for the reviewers and the community to know the limitations so that there could be more work/discussions in that direction to resolve it.
4. Minimal video results. For a paper proposing video generation, I would expect a minimum of 15-20 videos to be shown in the supplementary. It is difficult for the reviewer to judge if the results are cherry picked or does it actually generalise well on many videos and different camera trajectory.
5. There are no results shown in the case where given an initial video and 2-3 different trajectories, are all 2-3 videos coherent to the initial video atleast in the region of static part. Are the generated videos coherent with each other in the common regions?
6. What about a comparison against the newly released ReCamMaster? It is an ICCV paper I assume and hence the evaluation code has been released long before ICLR deadline. I would request the authors to compare this work with their method and show few comparison.

**Questions:**

I have highlighted the major questions in the weaknesses already, and I am summing them up below(I have summarised them shortly so that it is easier for the reviewer to quote the exact question they are answering. Please refer to the Weakness for the detailed problem and question asked.
1. Why is the spatial structure not preserved in the first supplementary video where two chairs were supposed to be side-by-side, but instead a white pillar or cupboard appears?
2. Is there any error in camera pose estimation that, when back-projected to image space, causes inaccurate projections and leads to inconsistency?
3. Are the comparisons fair, given that your method takes both the initial few frames and the last few frames as input, while the other methods only use the last frame as prior?
4. Are the better qualitative results mainly due to conditioning on more input frames rather than the proposed architectural changes?
5. Why didn’t the authors train a baseline like CameraCtrl or another strong model with minimal modifications to take more input frames and verify whether improvements come from more inputs or architecture changes?
6. What are the limitations of the proposed work?
7. What are the potential future directions or improvements for this work?
8. Why are there so few video results in the supplementary, only a handful instead of 15–20 videos as expected for a video generation paper?
9. Are the generated videos coherent with each other when given the same initial video but different trajectories, especially in the static regions?
10. Why hasn’t the paper compared its method with ReCamMaster, given that its evaluation code has been available since before the ICLR deadline?

---

> ### Author Response · Authors · 2025-11-26
> **Response to Reviewer xUs1 (1 / 4)**
>
> ### **Response to Weakness 1 and Question 1 regarding scene-consistency**
>
> ---
>
> We thank the reviewer for pointing this out. We observed that when using the default CFG scale (CFG = 6) of CogVideoX originally designed to effectively follow the text prompt, some of the generations were more strongly influenced by the text prompt than by the spatial conditioning, which was mainly due to an overly large classifier-free guidance (CFG) scale. This occasionally caused the model to prioritize semantic guidance over following geometric constraints, as in the first video, the reviewer has pointed out.
>
> To address this issue, we re-generated the affected examples using an appropriately calibrated CFG scale (CFG = 3) and added additional qualitative results in the supplementary materials. These updated outputs show that, with proper guidance balancing, our method reliably adheres to both the spatial conditioning and the intended camera trajectory while maintaining high visual fidelity.
>
> ---
>
> ### **Response to Weakness 4 and Question 8 regarding limited number of videos in supplementary materials**
>
> ---
>
> We apologize for the limited number of videos in the supplementary materials. We have added more results and provide a total of 19 videos in the updated video. We thank the reviewer for this constructive suggestion and would grateful if the reviewer could take a look at our updated video.
>
> ---
>
> ### **Response to Weakness 3 and Question 6 regarding missing limitations**
>
> ---
>
> We apologize for not including the limitations of our approach in our initial manuscript, and we fully agree with the reviewer that mentioning these limitations is extremely important for the research community. We have included a limitations section in Appendix D of our updated manuscript.
>
> ---
>
> ### **Response to Question 2**
>
> ---
>
> > Is there any error in camera pose estimation that, when back-projected to image space, causes inaccurate projections and leads to inconsistency?
> >
>
> ---
>
> As the reviewer has pointed out, one of the potential failure modes of our model would be the erroneous estimation of geometry from the SLAM component. Our method relies on warped static regions computed from SLAM-estimated poses; thus, when SLAM experiences drift or partial failures, these pose errors can propagate into the spatial conditioning. In such cases, the warped regions may become slightly misaligned with the intended camera path, which can weaken the geometric consistency of the final video. However, modern SLAM systems generally provide robust and reliable performance for typical in-the-wild scenarios, making such failures relatively rare. In addition, one of the advantages of our framework is that we do not have any special architecture designs tailored to MegaSAM and can always replace MegaSAM with a more robust and powerful model to resolve the current limitations. Here, we show an experiment where we replace MegaSAM with the recently released DepthAnything v3 for inference. The results, summarized in the table below, show clear and consistent improvements across all metrics, together with reduced inference time. This confirms that our method directly benefits from stronger priors and highlights a key advantage of our design: the components can be swapped without changing the overall pipeline, allowing users to flexibly choose different pretrained models.
>
> |  | PSNR ($\uparrow$﻿) | SSIM ($\uparrow$) | LPIPS ($\downarrow$) | Inference Time ($\downarrow$) |
> | --- | --- | --- | --- | --- |
> | Ours - MegaSAM | 13.0468 | 0.3666 | 0.3812 | 9 min 21.31 sec |
> | Ours - DepthAnything v3 | 13.4534 | 0.3980 | 0.3637  | 5 min 12.53 sec |

---

> ### Author Response · Authors · 2025-11-26
> **Response to Reviewer xUs1 (2 / 4)**
>
> ### **Response to Weakness 2 and Question 3 and 4 regarding fairness**
>
> ---
>
> > I feel like the comparisons are not fair. The video comparisons are all against methods which only do camera control video generation with only the last frame as prior.
> >
>
> > Are the comparisons fair, given that your method takes both the initial few frames and the last few frames as input, while the other methods only use the last frame as prior?
> >
>
> > Are the better qualitative results mainly due to conditioning on more input frames rather than the proposed architectural changes?
> >
>
> ---
>
> We appreciate the reviewer for raising this issue. As mentioned by the reviewer, to the best of our knowledge, there are no existing works with publicly available implementations that can solve the same task. Therefore, as mentioned in “Experimental Settings” in Section 4.1 of our main paper, we originally evaluated our method by decomposing our task into subproblems: 1. camera-controllability and 2. scene-consistency, where each problem was compared with prior baselines specialized for the task, where DFoT in Table 1 also takes multiple frames as input. Nevertheless, we agree with the reviewer that comparing with a baseline method to isolate the benefits of our proposed methods 1. last single frame → last 9 frames for improved motion smoothness, 2. spatial prompts for scene-consistency and precise camera control will better highlight the benefits of our proposed architecture.
>
> As extending or modifying previous I2V camera-controllable methods to take multiple frames as input is non-trivial, we show additional comparisons with multiple variants of our framework, which acts as an effective baseline. First, we isolate the benefits of providing last w = 9 frames, which is for enhanced motion smoothness by comparing with a I2V variant (w = 1) of our framework. To this end, we have added a new comparison using the w=1 (single-frame) setting shown in the following table, where we compare camera-controllability (mRotErr, mTransErr, mCamMC) and dynamic smoothness quality categories from VBench (Temporal Flicker, Motion Smooth) in the subset of the DynPose-100K dataset.
>
> |  | mRotErr ($\downarrow$) | mTransErr ($\downarrow$) | mCamMC ($\downarrow$) |
> | --- | --- | --- | --- |
> | MotionCtrl | 3.5654 | 7.8231 | 9.7834 |
> | CameraCtrl | 3.3273 | 9.5989 | 11.2122 |
> | FloVD | 3.4811 | 11.03032 | 12.6202 |
> | w=1 | 2.3898 | 7.7819 | 8.9785 |
> | w=9 (Ours) | 2.3772 | 7.4174 | 8.6352 |
>
> |  | Temporal Flicker ($\uparrow$) | Motion Smooth ($\uparrow$) |
> | --- | --- | --- |
> | w=1 | 0.8379 | 0.9253 |
> | w=9 (Ours) | 0.8561 | 0.9335 |
>
> As shown in the results, the results of w=1 and w=9 show similar performance in camera-controllability, while outperforming prior baseline works. However, aligned with our intentions where providing multiple frames leads to better motion smoothness, providing only a single frame leads to degraded performance in the VBench metrics. This added experiment isolates the benefits of our proposed spatial prompt as an effective cue for camera control and now added to Table 2 of our main paper and Table 7 of the Appendix.
>
> We further isolate the benefits of incorporating spatial prompts, which is the projection of the static point clouds lifted from the selected key frames based on spatial adjacency. In our experiments, the setting n=0, where n is the number of projected frames, corresponds to the baseline where the model is conditioned only on the temporal frames given as condition without any projected spatial views. The table is as follows, where we evaluate the scene-consistency (PSNR, SSIM, LPIPS) for trajectories that re-visit the camera poses in the given video and camera-controllability (mRotErr, mTransErr, mCamMC) in the DynPose-100K dataset:
>
> |  | PSNR ($\uparrow$﻿) | SSIM ($\uparrow$﻿) | LPIPS ($\downarrow$) | mRotErr ($\downarrow$) | mTransErr ($\downarrow$) | mCamMC ($\downarrow$) |
> | --- | --- | --- | --- | --- | --- | --- |
> | n=0, w=9 | 11.9555 | 0.3370 | 0.4512 | 3.4142 | 8.5920 | 10.5049 |
> | n=4, w=9 | 13.0382 | 0.3733 | 0.3758 | 2.3739 | 7.4278 | 8.6488 |
> | n=7, w=9 (Ours) | 13.0468 | 0.3666 | 0.3812 | 2.3772 | 7.4174 | 8.6352 |
>
> As shown in the table, without the incorporation of our spatial prompts, the baseline framework fails to maintain scene-consistency and fail to follow the camera condition. This effectively verifies the importance of our spatial prompts for both scene-consistency and precise camera control, as n=4 and n=7 significantly outperforms n=0 variant in all metrics. These results are now added to Table 8 of the Appendix.
>
> By combining the dual spatio-temporal conditioning, our framework enables scene-consistent camera-controllable video generation without introducing any architectural modifications of the pretrained DiT-based I2V video generation model. We appreciate the reviewer for this suggestion, which better highlights the importance and effectiveness of each our conditioning mechanisms.

---

> ### Author Response · Authors · 2025-11-26
> **Response to Reviewer xUs1 (3 / 4)**
>
> ### **Response to Weakness 6 and Question 10**
>
> ---
>
> > What about a comparison against the newly released ReCamMaster? It is an ICCV paper I assume and hence the evaluation code has been released long before ICLR deadline. I would request the authors to compare this work with their method and show few comparison.
> >
>
> > Why hasn’t the paper compared its method with ReCamMaster, given that its evaluation code has been available since before the ICLR deadline?
> >
>
> ---
>
> We thank the reviewer for this question. We would like to clarify the difference between the task of ReCamMaster and ours. As mentioned in our related works, ReCamMaster tackles the problem of “Dynamic Novel View Synthesis”, where they aim to generate novel views of a given video, within the same timestamps. Specifically, given a video from t=0 ~ t=N, they generate a video from a novel view but within the same temporal range of t=0 ~ t=N. In contrast, our goal is to generate the next-chunk(future video) of the given video while following the user-desired camera trajectory and maintaining scene-consistency with the given video. When provided with the same video from t=0 ~ t=N, our method aims to generate videos from t=(N+1) and after. This distinction in the task introduces unique and challenging problems, such as handling dynamic regions so that only the static regions remain consistent. Due to these differences, the output of ReCamMaster is the video within t=0 ~ t=N and the output of our framework becomes the video from t=(N+1), which is why we could not compare with ReCamMaster.
>
> ---
>
> ### **Response to Weakness 5 and Question 9**
>
> ---
>
> > There are no results shown in the case where given an initial video and 2-3 different trajectories, are all 2-3 videos coherent to the initial video atleast in the region of static part. Are the generated videos coherent with each other in the common regions?
> >
>
> > Are the generated videos coherent with each other when given the same initial video but different trajectories, especially in the static regions?
> >
>
> ---
>
> As mentioned in our previous response for **[Weakness 6, Question10]**, we focus on the task of generating the next-chunk (t = (N + 1) ~) of the given video (t = 0 ~ N), while following the user-desired camera trajectory and maintaining scene-consistency with the given video. When generating next-chunk videos with various trajectories, our main goal is to maintain scene-consistency of the static regions, which have been observed in the original input video. We achieve this by conditioning with spatial prompts, which are the projections of the static regions from the selected key frames from the input video based on spatial adjacency. On the other hand, as we are generating future videos, we do not have any regularizations to maintain consistency of the static regions that are newly explored in the timeframe of t =(N + 1) and after for videos that are generated in different generation processes (Video1, Video2). For these regions, our framework generates plausible content based on the given text prompt and the original scene for every generation process.

---

> ### Author Response · Authors · 2025-11-26
> **Response to Reviewer xUs1 (4 / 4)**
>
> ### **Response to Question 7**
>
> ---
>
> > What are the potential future directions or improvements for this work?
> >
>
> ---
>
> Although our primary interest is building a framework capable of generating a spatially consistent next-video chunk given an arbitrary video as context, by iteratively applying our method, one of the potential applications of our framework is generating videos of arbitrary length. To showcase this capability, we additionally conduct quantitative evaluations on long-video generation using the DAVIS dataset. We implement our iterative autoregressive pipeline by providing the last 9 frames from the previously generated video as temporal context, and select key frames for the spatial prompts from the entire previously generated video. We compare our framework against previous I2V baselines, where we apply a similar autoregressive approach. In this experiment, we evaluate video generation quality (PSNR, SSIM, LPIPS) and camera-controllability (RMSE, MSE, ATE).
>
> | Methods | PSNR ($\uparrow$) | SSIM ($\uparrow$) | LPIPS ($\downarrow$) | RMSE ($\downarrow$) | MSE ($\downarrow$) | ATE ($\downarrow$) |
> | --- | --- | --- | --- | --- | --- | --- |
> | CameraCtrl | 8.64 | 0.19 | 0.66 | 95.45 | 9332.08 | 0.1709 |
> | FloVD | 10.77 | 0.42 | 0.55 | 76.08 | 6131.95 | 0.2242 |
> | Ours | **17.28** | **0.60** | **0.35** | **37.28** | **1583.77** | **0.1794** |
>
> As shown in the results, our method shows its potential for long video generation, significantly outperforming previous baselines. For a detailed description of the experimental setup, please refer to Section A of the supplementary materials. These results are now added in Table 5 in the supplementary materials. We also provide additional long-video qualitative results in Figure 14 of the Appendix and Supplementary materials.
>
> However, as is common in autoregressive generation, extending our framework to extremely long durations introduces the challenge of temporal error accumulation. While our method provides the critical ***spatial consistency*** required for long videos, solving long-term drift (error accumulation) typically requires dedicated refinement modules and techniques. We believe that further extending our framework to mitigate the aforementioned issues is a very important and interesting future direction to explore.
>
> ---
>
> ### **Response to Question 5**
>
> ---
>
> > Why didn’t the authors train a baseline like CameraCtrl or another strong model with minimal modifications to take more input frames and verify whether improvements come from more inputs or architecture changes?
> >
>
> ---
>
> We thank the reviewer for this insightful question. We address this from two perspectives:
>
> (i) Technical Perspective: The performance gap stems fundamentally from the nature of the conditioning signal, rather than just the number of input frames. As demonstrated in Table 2, our explicit geometric conditioning significantly outperforms the camera controllability of CameraCtrl and MotionCtrl. We attribute this to a critical limitation in prior mechanisms: methods like CameraCtrl rely on **Plücker embeddings** or raw camera extrinsics, which suffer from **train-inference scale ambiguity**. These representations force the model to implicitly "learn" and memorize the specific metric scales observed during training, causing it to struggle when generalizing to new trajectories with different scales. In contrast, our approach projects static points directly from keyframes. This provides **explicit geometric guidance**, effectively removing the ambiguous step of "scale understanding" from the model. Therefore, simply increasing the context length of a Plücker-based model would not resolve this fundamental ambiguity, whereas our projection-based spatial prompts provide robust and precise control.
>
> (ii) Practical Perspective:  As mentioned by the reviewer, modifying an existing strong baseline is a straightforward starting point we also considered. However, we chose CogVideoX-I2V as our backbone due to significant architectural advantages. CameraCtrl and MotionCtrl rely on SVD (Stable Video Diffusion), which is a U-Net-based architecture. Recent literature demonstrates that DiT (Diffusion Transformer) models, such as CogVideoX, consistently outperform SVD-based models in generation quality. While other baselines like FloVD use CogVideoX, they introduce complexity by requiring two separate models (one for optical flow, one for generation). By building on CogVideoX directly, we achieve state-of-the-art quality with a simpler, unified architecture.

---

> ### Comment · Reviewer_xUs1 · 2025-11-28
> **Reply to authors**
>
> I have read all the clarifications given by the authors. I see that the authors have provided an exhaustive qualitative and quantitative analysis, and especially the video, which contains additional videos that prove the generalizability of the method. I really appreciate the author's effort to clarify Weakness 2 and Questions 3 and 4 in a systematic and quantitative manner. I am happy with the clarifications given by the authors.

---

> > ### Author Response · Authors · 2025-11-28
> >
> > Thank you for the positive evaluation of our work and for raising the score to 8! We also thank the reviewer for taking the time to provide a thorough review. Your feedback helps us a lot in improving our work!
> >
> > If you have any further questions, we’d be happy to discuss them during the discussion period.

---

### Official Review · Reviewer_iskH · 2025-10-28

**Soundness:** 2
**Presentation:** 3
**Contribution:** 2
**Rating:** 4
**Confidence:** 3

**Summary:**

The paper proposes a method to maintain geometric and temporal consistency for camera-controlled video extension generation. For temporal consistency, the last 9 frames of the original video are given to the model as a condition. For spatial consistency, the method extracts a 3D point cloud representation of the static scene from the original video and projects the point cloud to corresponding views, and provides the model as a condition, hence the name ScenePrompting. The 3D point cloud only encodes the static scene by masking out the dynamic objects. The paper's evaluation shows improved performance compared to prior methods, and the ablation studies validate the effectiveness of the design choice.

**Strengths:**

The paper is clearly written. The idea of using explicit 3D point cloud for long-term video generation memory is sound.

**Weaknesses:**

1. Missing comparisons to competing works [1, 2, 3]. The paper completely does not mention or compare with these prior works that also utilize explicit 3D memory for video generation.

2. Missing related works: CameraCtrl2 [4] explores improved camera conditioning and training on a large-scale foundational model. APT2 [5] explores long-duration and real-time camera-controlled generation. These works can be included in the related works section.

3. The paper proposes methods to remove dynamic objects from video. However, erasing and inpainting dynamic objects from video is an established research topic [5]. The paper does not mention such previous work.

4. The model in the paper can only generate short videos of a few seconds, which limits the actual use case of long-duration memory. The videos in the supplementary materials do not show significant improvement. Also, the method only compares against old works (CameraCtrl1 instead of CameraCtrl2).

[1] Video World Models with Long-term Spatial Memory

[2] Learning 3D Persistent Embodied World Models

[3] WORLDMEM: Long-term Consistent World Simulation with Memory

[4] CameraCtrl II: Dynamic Scene Exploration via Camera-controlled Video Diffusion Models

[5] Autoregressive Adversarial Post-Training for Real-Time Interactive Video Generation

[6] Flow-edge Guided Video Completion

**Questions:**

1. Regarding weakness 1, can the authors explain why competing explicit 3D memory works are not mentioned and compared against? I think this is critical for the paper's acceptance.

2. Regarding weakness 2, can the author explain the relationship between the proposed dynamic object removal pipeline and existing works?

3. The paper's introduction section inspires the development of the proposed method for long-duration generation. Can the author clarify the maximum duration the model supports? And also, how long were the generated videos used for evaluation? The supplementary material only shows very short example videos, where the original videos are so short and can completely be fitted to the attention as a condition, which eliminates the need for such complex spatial prompting design, and the generated videos are also short, which does not showcase the enhancement using the method.

---

> ### Author Response · Authors · 2025-11-26
> **Response to Reviewer iskH (1 / 3)**
>
> ### **Response to Weakness 1 and Question 1**
>
> ---
>
> > Missing comparisons to competing works [1, 2, 3]. The paper completely does not mention or compare with these prior works that also utilize explicit 3D memory for video generation.
> >
>
> > Regarding weakness 1, can the authors explain why competing explicit 3D memory works are not mentioned and compared against? I think this is critical for the paper's acceptance.
> >
>
> [1] Video World Models with Long-term Spatial Memory (to appear at NeurIPS Dec, 2025)
>
> [2] Learning 3D Persistent Embodied World Models (to appear at NeurIPS Dec, 2025)
>
> [3] WORLDMEM: Long-term Consistent World Simulation with Memory (to appear at NeurIPS Dec, 2025)
>
> ---
>
> We thank the reviewer for pointing out these relevant works. We would like to first mention that all three works [1,2,3] is a concurrent work following the ICLR policy and especially SPMem [1] which was arxived within 3 months of the ICLR submission deadline:
>
> ```
> ... "if a paper was published (i.e., at a peer-reviewed venue) on or after July 24, 2025, authors are not required to compare their own work to that paper. Note that arXiv is not considered a peer-reviewed venue." ...
> ```
>
> While we acknowledge the similarity of the mentioned works [1,2,3], we would like to clarify the difference of our work and why we did not include a direct comparison with these works.
>
> For [2], although they introduce a 3D memory, they propose an implicit memory and target **static scenes**, where our main task and contribution lie in leveraging explicit 3D memory for scene-consistent video generation in **real-world dynamic videos.** We analyze that when adopting 3D memory in dynamic scenes, it is important to handle dynamic regions, where we propose a multi-stage pipeline to identify and mask out these dynamic regions.
>
> WorldMem [3] is built upon Oasis [A], a diffusion forcing architecture trained solely on the Minecraft dataset, where they primarily address **static scenes (RealEstate10K)** or **synthetic domains (Minecraft)**. In addition, they additionally provide key frames as a condition by increasing the sequence size, which also restricts the model to take sufficient number of frames as condition when provided with long videos. In contrast, we focus on how we can leverage the learned priors of DiTs to generate scene-consistent camera-controllable videos in **complex real-world dynamic scenarios.** In addition, by projecting the constructed static-only 3D point clouds to the target trajectory, we can condition on long videos without increasing computation for the generation process.
>
> [1], which is a **concurrent** work regarding the ICLR submission date, is the most conceptually similar work to ours. However, it differs in two key aspects:
>
> 1. Dynamic object handling: While [1] computes the static component via TSDF fusion, we introduce more concise and accurate dynamic mask generation methods to remove dynamic objects.
> 2. Efficiency: Architecturally, [1] requires additional ControlNet-style Diffusion-as-Shader [B] architecture to incorporate the condition, which requires additional architecture search of the ControlNet-style block for different DiTs and require additional computational resources for training. On the other hand, we achieve scene-consistent camera-controllable generation ***without*** any architectural changes or additional modules. We simply replace the zero-paddings of the original I2V input by injecting spatial and temporal conditioning into the zero-padding slots, which enables efficient training and inference without introducing additional trainable modules.
>
> In addition, they do not have a publicly available code, which does not allow us to perform a direct comparison in performance. To clarify the difference of our work with [1,2,3], following the reviewer's suggestion, we included a detailed discussion and comparison with these relevant works in Section C of the Appendix of the revised manuscript to better emphasize our contributions. In addition, as our framework does not include any complex implementations, all the code will be publicly released.
>
> ---
>
> [A] Oasis: Decart, Etched, et al. "Oasis: A universe in a transformer." URL: [https://oasis-model](https://oasis-model/). github. io (2024).
>
> [B] Diffusion-as-Shader : Gu, Zekai, et al. "Diffusion as shader: 3d-aware video diffusion for versatile video generation control." Proceedings of the Special Interest Group on Computer Graphics and Interactive Techniques Conference Conference Papers. 2025.

---

> ### Author Response · Authors · 2025-11-26
> **Response to Reviewer iskH (2 / 3)**
>
> ### **Response to Weakness 2**
>
> ---
>
> > Missing related works: CameraCtrl2 [4] explores improved camera conditioning and training on a large-scale foundational model. APT2 [5] explores long-duration and real-time camera-controlled generation. These works can be included in the related works section.
> >
>
> [4] CameraCtrl II: Dynamic Scene Exploration via Camera-controlled Video Diffusion Models
>
> [5] Autoregressive Adversarial Post-Training for Real-Time Interactive Video Generation
>
> ---
>
> We apologize for missing to properly mention CameraCtrl2 and APT2 in the original submission. CameraCtrl2 employs Plücker embedding–based camera conditioning and uses the previously generated frames as input for extending videos, while APT2 also generates videos autoregressively using Plücker embeddings. Both methods explore camera-controllable generation, but they autoregressively generate videos by only taking the previous images as conditions, without modeling 3D scene structure or considering spatial adjacency. Therefore, these models fail to incorporate all previous frames due to computational constraints.
>
> In contrast, our approach performs precise camera control and scene-consistent video generation by constructing an explicit 3D static memory using point clouds. By projecting the static memory, this enables pixel-wise spatial alignment across views. Instead of relying on autoregressive propagation or feature-level Plücker conditioning, our 3D memory is directly integrated into the diffusion transformer via 3D-warped point clouds, providing scene-consistent constraints that do not accumulate drift over time. This design also allows us to handle dynamic objects by separating static and dynamic components using our dynamic mask, which is not addressed in these prior works.
>
> We have added this discussion in the related works and Appendix C section to the updated version. Although we would like to include direct comparisons, their implementations are not publicly available, making both qualitative and quantitative evaluations difficult.
>
> ---
>
> ### **Response to Weakness 3 and Question 2**
>
> ---
>
> > The paper proposes methods to remove dynamic objects from video. However, erasing and inpainting dynamic objects from video is an established research topic [5(6?)]. The paper does not mention such previous work.
> >
>
> > Regarding weakness 2(3?), can the author explain the relationship between the proposed dynamic object removal pipeline and existing works?
> >
>
> Based on the context, we assume the reviewer has intended to reference 6 instead of 5 in Weakness3, and reference Weakness 3 instead of Weakness 2 in Question2. Our response is as follows:
>
> [6] Flow-edge Guided Video Completion
>
> ---
>
> Although the conditioning of our spatial prompts seems similar to existing inpainting works [6], our spatial prompts are used as a soft spatial guide rather than a hard constraint, allowing the diffusion model to fill holes and refine details while still respecting the geometric layout where reliable projections exist. This ensures that the diffusion model retains the flexibility to **generate dynamic details on top of this condition** while the final output naturally blends geometric consistency with realistic temporal dynamics where an qualitative example is shown in Figure 8 of the Appendix. This fundamentally differs from [6], which operates as a standard inpainting model and does not reason about dynamic motions beyond filling masked regions. We have incorporated a detailed discussion in Section C of the Appendix, explicitly highlighting these fundamental differences into the revised manuscript.

---

> ### Author Response · Authors · 2025-11-26
> **Response to Reviewer iskH (3 / 3)**
>
> ### **Response to Weakness 4 and Question 3 regarding the application of our framework to long video generation**
>
> ---
>
> > The model in the paper can only generate short videos of a few seconds, which limits the actual use case of long-duration memory. The videos in the supplementary materials do not show significant improvement. Also, the method only compares against old works (CameraCtrl1 instead of CameraCtrl2).
> >
>
> > The paper's introduction section inspires the development of the proposed method for long-duration generation. Can the author clarify the maximum duration the model supports? And also, how long were the generated videos used for evaluation? The supplementary material only shows very short example videos, where the original videos are so short and can completely be fitted to the attention as a condition, which eliminates the need for such complex spatial prompting design, and the generated videos are also short, which does not showcase the enhancement using the method.
> >
>
> ---
>
> We clarify that our primary interest is building a framework **“capable of generating a spatially consistent next-video chunk given an arbitrary video as context”** as mentioned in our abstract, introduction, and methodology sections. However, as the reviewer mentioned, by iteratively applying our method, one of the potential applications of our framework is generating videos of arbitrary length. To showcase this capability, we additionally conduct quantitative evaluations on long-video generation using the DAVIS dataset. We implement our iterative autoregressive pipeline by providing the last 9 frames from the previously generated video as temporal context, and select key frames for the spatial prompts from the entire previously generated video. We compare our framework against previous I2V baselines, where we apply a similar autoregressive approach. For CameraCtrl, we also compare with the autoregressive variant enhanced with training-free long video generation techniques for SVD, such as FreeLong [A].  In this experiment, we evaluate video generation quality (PSNR, SSIM, LPIPS) and camera-controllability (RMSE, MSE, ATE).
>
> | Methods | PSNR ($\uparrow$) | SSIM ($\uparrow$) | LPIPS ($\downarrow$) | RMSE ($\downarrow$) | MSE ($\downarrow$) | ATE ($\downarrow$) |
> | --- | --- | --- | --- | --- | --- | --- |
> | CameraCtrl | 8.64 | 0.19 | 0.66 | 95.45 | 9332.08 | 0.1709 |
> | CameraCtrl + latent interpolation | 14.12 | 0.49 | 0.45 | 53.50 | 3214.71 | 0.2225 |
> | CameraCtrl + FreeLong [A] | 13.46 | 0.44 | 0.48 | 55.84 | 3314.31 | 0.2147 |
> | FloVD | 10.77 | 0.42 | 0.55 | 76.08 | 6131.95 | 0.2242 |
> | Ours | **17.28** | **0.60** | **0.35** | **37.28** | **1583.77** | **0.1794** |
>
> As shown in the results, our method significantly outperforms these enhanced baselines. For a detailed description of the experimental setup, please refer to Section A of the Appendix. These results are now added in Table 5 of the Appendix. We also provide additional long-video qualitative results in Figure 14 of the Appendix and supplementary videos.
>
> However, as is common in autoregressive generation, extending our framework to extremely long durations introduces the challenge of temporal error accumulation. While our method provides the critical *spatial consistency* required for long videos, solving long-term drift (error accumulation) typically requires dedicated refinement modules and techniques such as Self-Forcing [B], which are orthogonal to our core contribution and considered out of scope for this work. Therefore, we focused our evaluation on validating the spatial consistency within a single generation window, which serves as the fundamental building block for longer sequences.
>
> As mentioned above, CameraCtrl2 was excluded because its official implementation and pre-trained weights are not publicly available.
>
> ---
>
> [A] Lu, Yu, et al. "Freelong: Training-free long video generation with spectralblend temporal attention." NeurIPS’24.
>
> [B] Huang, Xun, et al. "Self Forcing: Bridging the Train-Test Gap in Autoregressive Video Diffusion." arXiv preprint arXiv:2506.08009 (2025).

---

> > ### Comment · Reviewer_iskH · 2025-11-26
> >
> > Thank you for providing a detailed response.
> >
> > 1. I agree that some of the related works are concurrent work under ICLR's rule to exclude in comparison. The authors have included them as discussion in appendix. I think this is sufficient.
> > 2. The authors have also updated the publication to include CameraCtrl2 and APT2 as related works. I agree that CameraCtrl2 and APT2 has a different main objective and the authors' work focuses on spatial consistency. Since CameraCtrl2 and APT2 are not directly comparable. It is reasonable to not include as comparison.
> >
> > Remaining concern:
> > 1. The samples provided in the paper (Figure 6 and 7 in the main text as well as the new figure 8) all are very tiny camera movements. The new figure 8 shows t=40 long duration generation but the background merely moved. This is against the whole purpose of the method which is to allow scene consistency under large camera movement. For small camera movement, adjacent frame information is enough and there is no reason for the whole scene extraction process. This poorly reflect the practicality and applicability of the method.
> >
> > Can the paper showcase examples such as: completely turning the camera away and swing back to show that the scene is consistency?

---

> > > ### Author Response · Authors · 2025-12-01
> > > **Response to follow-up question by Reviewer iskH**
> > >
> > > As suggested by the reviewer, we have added more examples to the supplementary videos with trajectories that better showcase the scene-consistent generation capability of our approach. We would be grateful if the reviewer can take a look at the most recent supplementary video. Although the camera movement can be small, the video generative framework only takes the last 9 frames as a temporal condition, which often does not contain regions that were observed in the entire input video. Specifically, in the first example in the supplementary video, the input video has observed the kitchen area in the top-left region, but the 9 temporal frames do not contain this information. Also, in the third example, the background of the right vet in the surgery room is observed in the input video but not in the 9 temporal frames. We successfully generate scene-consistent videos by additionally leveraging the spatial memory, which projects static point clouds from key frames selected by spatial adjacency to compensate for regions that were observed in the input video but not in the temporally adjacent frames.

---

> ### Author Response · Authors · 2025-11-27
> **Response to follow-up question by Reviewer iskH**
>
> > The samples provided in the paper (Figure 6 and 7 in the main text as well as the new figure 8) all are very tiny camera movements. The new figure 8 shows t=40 long duration generation but the background merely moved. This is against the whole purpose of the method which is to allow scene consistency under large camera movement. For small camera movement, adjacent frame information is enough and there is no reason for the whole scene extraction process. This poorly reflect the practicality and applicability of the method.
> >
>
> We thank the reviewer for such a prompt review of our response and for the insightful suggestion. We agree that demonstrating results in the suggested scenarios will better highlight the advantages of our framework. We are currently generating examples to better showcase our framework in this specific setting to address your concern. We will update our response and the paper with these new results as soon as they are ready.

---

### Official Review · Reviewer_hP5a · 2025-10-30

**Soundness:** 3
**Presentation:** 3
**Contribution:** 3
**Rating:** 6
**Confidence:** 5

**Summary:**

This paper presents 3DScenePrompt, a framework for generating camera-controllable videos from arbitrary-length input videos while maintaining scene consistency. The key innovation is a dual spatio-temporal conditioning strategy: the model conditions on both temporally adjacent frames (last 9 frames) for motion continuity and spatially adjacent frames for scene consistency. To handle dynamic elements correctly, the authors construct a 3D scene memory using dynamic SLAM that represents only static geometry, extracted via a three-stage dynamic masking pipeline (pixel-level motion detection, SAM2 propagation, and backward tracking with CoTracker3). This static-only 3D representation is projected to target viewpoints to provide spatial prompts while allowing dynamics to evolve naturally. Experiments on RealEstate10K and DynPose-100K show improvements over baselines in scene consistency, camera control, and video quality.

**Strengths:**

1. The motivation of the paper is clear: when generating the videos consistent with previous frames, the static and dynamic element must be handled differently, and the underlying 3D geometry must be kept during the camera controlling.

2.  The dual spatio-temporal conditioning strategy is intuitive yet powerful. The recognition that spatial conditioning must provide only static geometry while temporal conditioning handles dynamics is key.

3. Building on frozen CogVideoX preserves pretrained priors and suggests the approach could generalize to other video models.

**Weaknesses:**

1. The ablation study is not sufficient, see the question parts.
2. There are some recent works try to address the scene-consistent camera controlled video generation (Line 336), for example, Cameractrl 2 and apt2.
3. The method only tested on CogVideoX.

**Questions:**

1. What happens when SLAM fails (texture-less scenes, fast motion, dynamic-dominated scenes)?
2. The paper shows only success cases. When does the method fail?
3. Table 4 only compares with/without the full mask. What's the contribution of each stage (pixel detection, SAM2, backward tracking)?
4. The temporal window size is fixed at 9 with no justification or ablation.
5. There should be a baseline method, that the model does not condition on the projected views, instead the original chosen temporal frames.
5. Does the method work on other video diffusion models beyond CogVideoX?

---

> ### Author Response · Authors · 2025-11-26
> **Response to Reviewer hP5a (1 / 3)**
>
> We thank the reviewer for the positive evaluation of our work and insightful feedback. Our detailed response can be seen below:
>
> ### **Response to Weakness 2**
>
> ---
>
> > There are some recent works try to address the scene-consistent camera controlled video generation (Line 336), for example, Cameractrl 2 and apt2.
> >
>
> ---
>
> We thank the reviewer for pointing out recent works such as CameraCtrl2 and APT2. We apologize for missing to properly reference CameraCtrl2 in our original paper, and we were not aware of the concurrent work Seaweed-APT2 at the time of our submission. In the revised manuscript, we have added a dedicated discussion in the Related Work section (see Multi-frame conditioned camera-controllable video generation paragraph) and Section C of the Appendix to position our method with respect to these approaches and to clarify the conceptual differences.
>
> While CameraCtrl2 and APT2 explores camera-controllable video generation, they do not strictly target ***“scene-consistency”.*** CameraCtrl2 employs Plücker embedding–based camera conditioning and uses the previously generated frames as input for extending videos, while APT2 also generates videos autoregressively using Plücker embeddings. Both methods only focus on the previous images as conditions, without modeling 3D scene structure or considering spatial adjacency. Therefore, these models fail to incorporate all previous frames due to computational constraints. In contrast, we mainly focus on how we can achieve ***“scene-consistency”*** without increasing the required computational resources, where we introduce an explicit 3D static memory using point clouds. By projecting point clouds from key frames selected by spatial adjacency, we achieve both scene-consistency and precise camera control enabling pixel-wise spatial alignment across views.
>
> Although we would like to include direct comparisons, their implementations are not publicly available, making both qualitative and quantitative evaluations difficult.
>
> ---
>
> ### **Response to Weakness 3 and Question 6**
>
> > The method only tested on CogVideoX.
> >
>
> > Does the method work on other video diffusion models beyond CogVideoX?
> >
>
> ---
>
> Our architecture is applicable to most DiT-based image-to-video (I2V) models, as we do not introduce any CogVideoX-specific designs. In conventional I2V pipelines, only the first-frame latent (image condition) is preserved, while the remaining latents are zero-padded and concatenated with the target noise along the channel dimension before entering the DiT. We simply replace the zero-paddings by injecting spatial and temporal conditioning into the zero-padding slots. Since we only modify the original zero-padding strategy typically used in I2V-DiT models, our method can be seamlessly integrated into essentially any DiT-based I2V architecture (e.g., Wan2.1). Although we wish to highlight the versatility of our framework by implementing it on top of Wan 2.1, due to largely different codebases of Wan 2.1 vs. CogVideoX and different model configurations (e.g., the number of frames generated, training epoch), we believe that completing this experiment within the rebuttal period is extremely challenging. Thus, although we cannot share the results during the rebuttal period, we will make sure to add the results of replacing CogVideoX with Wan2.1 in the final version of our manuscript.

---

> > ### Comment · Reviewer_hP5a · 2025-11-27
> >
> > I agree with that due to the large codebase gaps between CogVideoX and Wan, it is impossible for authors to implement the proposed methods on Wan or other open source video diffusion models. Hoping the author can add more results in the future version, and add some discussions with some concurrent works.

---

> ### Author Response · Authors · 2025-11-26
> **Response to Reviewer hP5a (2 / 3)**
>
> ### **Response to Question 1 and Question 2**
>
> ---
>
> > What happens when SLAM fails (texture-less scenes, fast motion, dynamic-dominated scenes)?
> >
>
> > The paper shows only success cases. When does the method fail?
> >
>
> ---
>
> As the reviewer has pointed out, one of the failure modes of our model would be the total failure of the SLAM component when the input video contains significantly rapid motion or when most of the input video consists of dynamic regions (e.g., selfie video). When the SLAM component totally fails to reconstruct geometry and provides no spatial condition to the model, our framework will switch back to the I2V generation, which will have little capability left to follow the conditioning trajectory. However, modern SLAM systems generally provide robust and reliable performance for typical in-the-wild scenarios, making such failures relatively rare. In addition, one of the advantages of our framework is that we do not have any special architecture designs tailored to MegaSAM and can always replace MegaSAM with a more robust and powerful model to resolve the current limitations. Here, we show an experiment where we replace MegaSAM with the recently released DepthAnything v3 for inference. The results, summarized in the table below, show clear and consistent improvements across all metrics, together with reduced inference time. This confirms that our method directly benefits from stronger priors and highlights a key advantage of our design: the components can be swapped without changing the overall pipeline, allowing users to flexibly choose different pretrained models.
>
> |  | PSNR ($\uparrow$﻿) | SSIM ($\uparrow$) | LPIPS ($\downarrow$) | Inference Time ($\downarrow$) |
> | --- | --- | --- | --- | --- |
> | Ours - MegaSAM | 13.0468 | 0.3666 | 0.3812 | 9 min 21.31 sec |
> | Ours - DepthAnything v3 | 13.4534 | 0.3980 | 0.3637  | 5 min 12.53 sec |
>
> This table is now added in Table 9 and Table 10 of the Appendix.
>
> ### **Response to Question 3**
>
> ---
>
> > Table 4 only compares with/without the full mask. What's the contribution of each stage (pixel detection, SAM2, backward tracking)?
> >
>
> ---
>
> We appreciate the reviewer for pointing this out. To better understand the contributions of each stage of our dynamic mask generation pipeline, we conduct a comprehensive ablation study, where we evaluate the video generation quality (PSNR, SSIM, LPIPS), scene-consistency (Met3r), and camera-controllability (mRotErr, mTransErr, mCamMC) in a subset of the DynPose-100K dataset. Here, we utilize the dynamic mask achieved from each stage of our dynamic mask generation pipeline: pixel detection $\rightarrow$ SAM2 detection $\rightarrow$ backward tracking (ours).
>
> |  | PSNR ($\uparrow$﻿) | SSIM ($\uparrow$﻿) | LPIPS ($\downarrow$) | Met3r ($\downarrow$) | mRotErr ($\downarrow$) | mTransErr ($\downarrow$) | mCamMC ($\downarrow$) |
> | --- | --- | --- | --- | --- | --- | --- | --- |
> | w/o dynamic mask | 11.9963 | 0.2898 | 0.3748 | 0.104002 | 3.4142 | 8.5920 | 10.5049 |
> | + pixel detection | 12.1083  | 0.3248  | 0.3690  | 0.155589 | 2.9390  | 7.7819  | 9.1524  |
> | + SAM2 detection | 13.5957  | 0.4036  | 0.3548  | 0.157610  | 2.7188  | 7.5402   | 8.9696  |
> | + backward tracking | 13.7311  | 0.4112  | 0.3454  | 0.122859 | 2.6103  | 7.4858  | 8.9181  |
>
> As shown in the above table, adding each of our proposed pipeline of pixel detection $\rightarrow$ SAM2 detection $\rightarrow$ backward tracking, leads to consistent improvement in all metrics, effectively verifying the contributions of each stage of our dynamic mask generation pipeline. This table is now added in Table 6 of the Appendix.

---

> > ### Comment · Reviewer_hP5a · 2025-11-27
> >
> > The rebuttal of this part has addressed most of my concern. There is an additional question: The model conditions on camera trajectory (Equ. 4) when generating each video chunk, thus what is the necessity of using SLAM to extract the camera trajectory of generated video chunk? Why not directly use the condition camera trajectory?

---

> > > ### Author Response · Authors · 2025-11-27
> > > **Response to follow-up question by Reviewer hP5a**
> > >
> > > > The model conditions on camera trajectory (Equ. 4) when generating each video chunk, thus what is the necessity of using SLAM to extract the camera trajectory of generated video chunk? Why not directly use the condition camera trajectory?
> > > >
> > >
> > > We thank the reviewer for the quick response and for this question. Regarding Equation 4, the provided camera trajectory $\mathbf{C}$ to the video generation model can indeed enable us to directly condition the model with this trajectory using Plücker embeddings as done in prior I2V camera-controllable video generation models (CameraCtrl, MotionCtrl) as the reviewer mentioned. We would like to explain the motivation of using SLAM in two perspectives:
> > >
> > > **(i) Camera-Controllability:** When using the Plücker embeddings or the camera extrinsic parameters directly as condition, they often suffer from **train-inference scene scale ambiguity**. While scene scale is different for different scenes, these conditioning mechanisms provide only the camera parameters, which require the model to implicitly "learn" and memorize the specific metric scales observed during training. This often leads to degraded camera-controllability to new trajectories with different scales. In contrast, by applying SLAM to the input video, we can understand the relative positions of the frames in the input video with the camera trajectory we want to generate. And instead of only providing camera parameters for camera control, our approach projects static points directly from the selected keyframes. This provides **explicit geometric guidance**, effectively removing the ambiguous step of "scale understanding" from the model, as we already provide the information on how the static points should be warped. We observed that our proposed conditioning achieves more precise camera control than previous methods in Table 2. We believe that one potential future direction would be combining the conditioning mechanisms of both our static point projection and direct camera trajectory conditionings.
> > >
> > > **(ii) Scene-Consistent Video Generation:** Given our main task of scene-consistent camera-controllable video generation, one of the key challenges is to maintain scene-consistency with the input video. Although a straightforward solution would be to condition the framework with all frames, this is infeasible due to computational constraints. Instead of conditioning on all frames, we propose the dual spatio-temporal conditioning mechanism, where we identify spatially adjacent frames from the input video with the given conditioning trajectory $\mathbf{C}$. To efficiently identify these key frames, we adopt SLAM to the input video, where we project the static point clouds from the selected key frames for both scene-consistency and better camera-controllability.

---

> > > > ### Comment · Reviewer_hP5a · 2025-11-27
> > > >
> > > > Regarding the scale:
> > > > 1. Are the samples in the training set of the same scale? e.g. metric scale
> > > > 2. Basically, the SLAM method estimates the camera parameters in relative scale, meaning running SLAM for the same video chunk two times may results in different scales (reflect in different camera distance in consecutive frames), or different scales for two consecutive video chunks, (e.g. the 1st chunk is in metric scale, while the 2nd chunk is in half metric scale).
> > > > Thus, the question is: how does the author postprocess the SLAM-estimated camera trajectories such that different chunks have the same scale?

---

> > > > > ### Author Response · Authors · 2025-11-27
> > > > > **Response to follow-up question by Reviewer hP5a**
> > > > >
> > > > > For the 1st question, as most datasets process data with the same algorithm or the same SLAM or SfM method to obtain camera poses, they are often in the same scale but not always metric. This leads to difficulties when providing arbitrary trajectories for inference, as the user is not aware of the scale the model was learned, leading to imperfect camera-controllability. For the 2nd question, we would like to first clarify that we do not have two video chunks; we have a single, arbitrary-length input video where we generate the next chunk. For evaluation with previous I2V camera-controllable video generation methods, we evaluate their method by providing the ground-truth camera trajectories in the dataset. For our framework, by applying SLAM to the input video, we obtain the camera poses for the input video. To generate videos following desired camera trajectories, the most straightforward way is to leverage an interactive viewer with the reconstructed scene, where the user can provide camera trajectories that directly match the scale. With only camera trajectories, we can further match the scales using the Procrustes and Umeyama normalization, leveraging the overlapping frames provided as temporal context. After normalization, as mentioned in our previous response, we provide conditioning by directly warping the point clouds, removing the ambiguous step of "scale understanding" from the model. For our additional experiments highlighting the potential application for scene-consistent **long** video generation, we iteratively applied our whole framework, where we applied SLAM iteratively on the newly concatenated video, while maintaining the already identified dynamic regions. We believe that further improving the framework for long video generation is an interesting future direction.

---

> > > > > > ### Comment · Reviewer_hP5a · 2025-11-27
> > > > > >
> > > > > > I have no more questions, and tend to accept this paper.

---

> > > > > > > ### Author Response · Authors · 2025-11-28
> > > > > > >
> > > > > > > Thank you very much for taking the time to provide a thorough review and positive evaluation of our work. Your feedback helps us a lot in improving our work!
> > > > > > >
> > > > > > > If you have any further questions, we’d be happy to discuss them during the discussion period.

---

> ### Author Response · Authors · 2025-11-26
> **Response to Reviewer hP5a (3 / 3)**
>
> ### **Response to Question 4**
>
> > The temporal window size is fixed at 9 with no justification or ablation.
> >
>
> ---
>
> As CogVideoX generates 49 frames at once, we can choose the temporal window size between 1 < w < 49, where (49-w) becomes the frames that are generated. In our framework, to effectively solve scene-consistent camera-controllable video generation, providing the last few frames (w=9) is one of our important architecture designs and contributions that enhance smooth dynamics and motion coherency with the original input video. We have empirically set w=9 after initial experiments that w=9 provides sufficient information to produce smooth motion continuity with the conditioning frames, while also generating 40 frames. To better understand the contribution of the temporal window size, we added experiments using the number of temporally adjacent frames (w) with w=1 and w=5. The table is as follows, where we evaluate the video camera-controllability (mRotErr, mTransErr, mCamMC) and motion smoothness (VBench - Temporal Flicker, Motion Smooth) in a subset of the DynPose-100K dataset:
>
> |  | mRotErr ($\downarrow$) | mTransErr ($\downarrow$) | mCamMC ($\downarrow$) | Temporal Flicker ($\uparrow$) | Motion Smooth ($\uparrow$) |
> | --- | --- | --- | --- | --- | --- |
> | w=1 | 2.3898 | 7.7819 | 8.9785 | 0.8379 | 0.9253 |
> | w=5 | 2.3837 | 7.5512 | 8.6233 | 0.8508 | 0.9262 |
> | w=9 (Ours) | 2.3772 | 7.4174 | 8.6352 | 0.8561 | 0.9335 |
>
> As shown in the table, the size of the temporal window does not have significant affect on camera-controllability, showing similar results. However, aligned with our intentions where providing the last few frames will improve motion smoothness and motion coherence, providing more frames lead to better VBench metrics. These results are now added to Table 7 of the Appendix.
>
> ---
>
> ### **Response to Question 5**
>
> > There should be a baseline method, that the model does not condition on the projected views, instead the original chosen temporal frames.
> >
>
> ---
>
> We appreciate the suggestion. We additionally evaluate the suggested baseline and report the results in the following table. In our experiments, the setting n=0, where n is the number of projected frames, corresponds to the baseline where the model is conditioned only on the temporal frames given as condition without any projected spatial views. The table is as follows, where we evaluate the scene-consistency (PSNR, SSIM, LPIPS) for trajectories that re-visit the camera poses in the given video and camera-controllability (mRotErr, mTransErr, mCamMC) in the DynPose-100K dataset:
>
> |  | PSNR ($\uparrow$﻿) | SSIM ($\uparrow$﻿) | LPIPS ($\downarrow$) | mRotErr ($\downarrow$) | mTransErr ($\downarrow$) | mCamMC ($\downarrow$) |
> | --- | --- | --- | --- | --- | --- | --- |
> | n=0, w=9 | 11.9555 | 0.3370 | 0.4512 | 3.4142 | 8.5920 | 10.5049 |
> | n=4, w=9 | 13.0382 | 0.3733 | 0.3758 | 2.3739 | 7.4278 | 8.6488 |
> | n=7, w=9 (Ours) | 13.0468 | 0.3666 | 0.3812 | 2.3772 | 7.4174 | 8.6352 |
>
> As shown in the table, without the incorporation of our spatial prompts, the baseline framework fails to maintain scene-consistency and fail to follow the camera condition. This effectively verifies the importance of our spatial prompts for both scene-consistency and precise camera control, as n=4 and n=7 significantly outperforms n=0 variant in all metrics. These results are now added to Table 8 of the Appendix.

---

> > ### Comment · Reviewer_hP5a · 2025-11-27
> >
> > The rebuttal of this part has addressed my concerns regarding more ablation and comparison.

---

### Official Review · Reviewer_3iY4 · 2025-10-31

**Soundness:** 2
**Presentation:** 3
**Contribution:** 3
**Rating:** 4
**Confidence:** 3

**Summary:**

The paper proposes 3DScenePrompt, a framework for scene-consistent, camera-controllable video generation from an arbitrary-length input video. The core idea is a dual spatio-temporal conditioning scheme: (i) a short temporal window of recent frames to preserve motion continuity, and (ii) spatial window obtained by projecting a static-only 3D scene memory built from the entire input via dynamic SLAM + dynamic masking. The static memory is rendered to the target viewpoints and concatenated with the temporal window to condition a CogVideoX-I2V backbone. Experiments on RealEstate10K and DynPose-100K report improved spatial/geometric consistency, camera controllability, and overall video quality over baselines including DFoT and recent camera-control methods.

**Strengths:**

- Dual spatio-temporal conditioning: retrieving spatially adjacent content by pose similarity, not just the most recent frames.

- Static-only 3D scene memory: A practical masking pipeline (flow residuals -> backward point tracking -> SAM2 object masks) to construct a static-only 3D scene memory by excluding dynamics before projecting to target viewpoints.

- Experimental results show that the method performs favorably against baselines. There is also a nice spread of different metrics: geometric consistency, spatial revisitation metrics, camera-following errors, FVD, and VBench.

- Ablations isolate the value of dynamic masking and the number of spatial frames.

**Weaknesses:**

- Fairness of controllability comparisons: Competing camera-control baselines are evaluated with their standard inputs (single image or text + camera), whereas your method consumes last 9 frames + spatial prompts, which is strictly richer conditioning. This can advantage camera trajectory adherence and quality metrics. Consider a variant that is restricted to the same conditioning budget (e.g., single frame + spatial prompts vs single frame) to isolate the benefit of spatial prompts rather than extra temporal frames.

- Running the entire pipeline seems computationally expensive. What is the end-to-end run time during inference compared to the baselines?

- The spatial retrieval is defined by FOV overlap to the planned camera. It’s less clear how the method performs when the planned trajectory explores previously unseen regions or when SLAM coverage is sparse/erroneous.

- How sensitive is the model w.r.t. the output errors/bias of the pretrained models? What are the failure modes? What happens when we use different pretrained models?

**Questions:**

In addition to the weakness section listed above, I have the following questions:

- How exactly is the pose similarity is computed and what are the thresholds?

- The paper argues benefits for consistency, but doesn’t show what happens when the planned camera explores regions poorly covered by SLAM. What happens there?

- How are colors/features assigned when projecting? How do you handle holes or depth ordering?

---

> ### Author Response · Authors · 2025-11-26
> **Response to Reviewer 3iY4 (1 / 3)**
>
> We sincerely appreciate your thoughtful feedback. Our detailed response can be seen below:
>
> ### **Response to Weakness 1 regarding fairness in camera-controllability evaluation**
>
> ---
>
> We appreciate the reviewer’s insightful suggestion. To effectively solve scene-consistent camera-controllable video generation, providing the last few frames (default 9 views in our framework) is one of our important architecture designs and contributions that enhance smooth dynamics and motion coherency with the original input video. We agree that conditioning on multiple views can improve adherence to the target camera trajectory and lead to higher-quality generated videos, and that comparing against models capable of multi-frame conditioning would provide a more comprehensive and fair evaluation. However, to the best of our knowledge, there are no publicly available implementations that support multi-frame inputs, which restricted us to compare against existing I2V camera-control baselines (CameraCtrl, MotionCtrl, and FloVD).
>
> We also agree with the reviewer that a comparison under a strictly fair setting, where only a single frame is provided for our model, would improve the completeness of the manuscript by isolating the advantages of multi-frame conditioning and incorporating our spatial prompts. To this end, we have added a new comparison using the w=1 (single-frame) setting shown in the following table, where we compare camera-controllability (mRotErr, mTransErr, mCamMC) and dynamic smoothness quality categories from VBench (Temporal Flicker, Motion Smooth) in the subset of the DynPose-100K dataset.
>
> |  | mRotErr ($\downarrow$) | mTransErr ($\downarrow$) | mCamMC ($\downarrow$) |
> | --- | --- | --- | --- |
> | MotionCtrl | 3.5654 | 7.8231 | 9.7834 |
> | CameraCtrl | 3.3273 | 9.5989 | 11.2122 |
> | FloVD | 3.4811 | 11.03032 | 12.6202 |
> | w=1 | 2.3898 | 7.7819 | 8.9785 |
> | w=9 (Ours) | 2.3772 | 7.4174 | 8.6352 |
>
> |  | Temporal Flicker ($\uparrow$) | Motion Smooth ($\uparrow$) |
> | --- | --- | --- |
> | w=1 | 0.8379 | 0.9253 |
> | w=9 (Ours) | 0.8561 | 0.9335 |
>
> As shown in the results, the results of w=1 and w=9 show similar performance in camera-controllability, while outperforming prior baseline works. However, aligned with our intentions where providing multiple frames leads to better motion smoothness, providing only a single frame leads to degraded performance in the VBench metrics. This added experiment isolates the benefits of our proposed spatial prompt as an effective cue for camera control and now added to Table 2 of our main paper and Table 7 of the Appendix.
>
> ### **Response to Weakness 2 regarding end-to-end computation cost**
>
> ---
>
> We thank the reviewer for the question. In the table below, we summarize the inference time and the underlying architecture leveraged by each method.
>
> |  | SLAM-Processing | Dynamic Masking & Depth Warping Time | Video Generation Time | Inference Time |
> | --- | --- | --- | --- | --- |
> | CameraCtrl | - | - | 1 min 38 sec | 1 min 38 sec |
> | MotionCtrl | - | - | 2 min 9.5 sec | 2 min 9.5 sec |
> | FloVD | - | - | 8 min 5.32 sec | 8 min 5.32 sec |
> | Ours - MegaSAM | 4 min 18.813 sec | 58.88 sec  | 4 min 3.62 sec  | 9 min 21.31 sec |
> | Ours - DepthAnything v3 | 10.031 sec  | 58.88 sec  | 4 min 3.62 sec  | 5 min 12.53 sec |
>
> From this comparison, we observe that the SLAM-processing time required by MegaSAM to construct our 3D memory adds 5 minutes to the end-to-end generation pipeline, resulting in slower generation times than existing methods. However, our main advantage lies in the construction and adaptation of the 3D memory, rather than the choice of the off-the-shelf model MegaSAM, enabling us to replace MegaSAM with a more advanced and efficient method to accelerate inference. Here, we also show the inference time of our framework when replacing MegaSAM with a recently released DepthAnything v3 model. By replacing MegaSAM with DepthAnything v3, we achieve substantial reduction in processing time and significantly faster results than FloVD, which leverages the same video diffusion backbone as ours (CogVideoX). In addition, the table below reports the results, showing that we obtain not only reduced inference time but also improved performance. We wish to highlight that our pipeline is inherently flexible, and we believe that the advances in the 4D reconstruction field will further enable our framework to improve performance and efficiency.
>
> |  | PSNR ($\uparrow$﻿) | SSIM ($\uparrow$) | LPIPS ($\downarrow$) |
> | --- | --- | --- | --- |
> | Ours - MegaSAM | 13.0468 | 0.3666 | 0.3812 |
> | Ours - DepthAnything v3 | 13.4534 | 0.3980 | 0.3637  |
>
> This table is now added in Table 9 and Table 10 of the Appendix.

---

> ### Author Response · Authors · 2025-11-26
> **Response to Reviewer 3iY4 (2 / 3)**
>
> ### **Response to Weakness3 and Question2 regarding sparse SLAM coverage**
>
> ---
>
> We highly appreciate the reviewer’s question. To better explain how our proposed method performs under trajectories that explore previously unseen regions, we would like to briefly revisit how our model processes the conditioning inputs. In conventional I2V pipelines, only the first-frame latent (image condition) is preserved, while the remaining latents are zero-padded and concatenated with the target noise along the channel dimension before entering the DiT. In our approach, we replace these zero-padded regions with our spatial prompts; however, pixels that do not receive a valid projection from the static point cloud remain black (zero-valued). Crucially, this means that for unseen regions, the input naturally reverts to the standard zero-padded configuration. As illustrated in the 2nd example of the supplementary video, when the planned path traverses these sparsely covered regions, our model respects the spatial conditioning where available and gracefully transitions to an image-to-video–style generation where it is not. In these SLAM-poor regions, the diffusion model relies on its learned priors to maintain visual realism and temporal coherence similar to the original I2V generation, ensuring that the output does not collapse but instead smoothly extrapolates beyond the reconstructed area.
>
> ### **Response to Weakness 4**
>
> ---
>
> > How sensitive is the model w.r.t. the output errors/bias of the pretrained models? What are the failure modes? What happens when we use different pretrained models?
> >
>
> We thank the reviewer for this question. As we introduce a versatile framework for scene-consistent camera-controllable video generation by combining an off-the-shelf SLAM module and a pretrained DiT-based video generation framework, we can replace the SLAM or DiT framework with different pretrained models. One of the failure modes of our model would be the total failure of the SLAM component when the input video contains significantly rapid motion or when most of the input video consists of dynamic regions (e.g., selfie video). When the SLAM component totally fails to reconstruct geometry and provides no spatial condition to the model, our framework will switch back to the I2V generation, which will have little capability left to follow the conditioning trajectory. However, one of the advantages of our framework is that we do not have any special architecture designs tailored to MegaSAM or CogVideoX, and can always be replaced with a more robust and powerful model to resolve the current limitations.
>
> For the SLAM component, we can replace MegaSAM with any other method that is capable of reconstructing dynamic scenes. Here, we show an experiment where we replace MegaSAM with the recently released DepthAnything v3 for inference. The results, summarized in the table below, show clear and consistent improvements across all metrics, together with reduced inference time. This confirms that our method directly benefits from stronger priors and highlights a key advantage of our design: the components can be swapped without changing the overall pipeline, allowing users to flexibly choose different pretrained models.
>
> |  | PSNR ($\uparrow$﻿) | SSIM ($\uparrow$) | LPIPS ($\downarrow$) | Inference Time ($\downarrow$) |
> | --- | --- | --- | --- | --- |
> | Ours - MegaSAM | 13.0468 | 0.3666 | 0.3812 | 9 min 21.31 sec |
> | Ours - DepthAnything v3 | 13.4534 | 0.3980 | 0.3637  | 5 min 12.53 sec |
>
> For the DiT component, as our framework does not introduce any CogVideoX-specific methods, our model is robust across different DiT-based I2V diffusion architectures and can be applied in the same manner to models such as Wan 2.1. Specifically, our approach modifies the zero-padding mechanism of I2V diffusion models by injecting spatial and temporal conditioning into the padding slots, which enables our framework to seamlessly integrate any DiT-based I2V backbone. Although we wish to highlight the versatility of our framework by implementing it on top of Wan 2.1, due to largely different codebases of Wan 2.1 vs. CogVideoX and different model configurations (e.g., the number of frames generated, training epoch), we believe that completing this experiment within the rebuttal period is extremely challenging. Thus, although we cannot share the results during the rebuttal period, we will make sure to add the results of replacing CogVideoX with Wan2.1 in the final version of our manuscript.

---

> ### Author Response · Authors · 2025-12-02
> **Response to Reviewer 3iY4 (3 / 3)**
>
> We apologize for missing out to clarify Q1, Q3 in our initial response. We found out these responses were not uploaded due to the length limitation and clarify these questions below.
>
> ### **Response to Question 1 and Question 3 regarding the details of spatial prompt conditioning**
> ---
> > How exactly is the pose similarity is computed and what are the thresholds?
> >
>
> > How are colors/features assigned when projecting? How do you handle holes or depth ordering?
> >
>
> We thank the reviewer for the detailed question. Pose similarity between the planned camera and SLAM keyframes is defined in terms of static-region FOV overlap. Concretely, for each planned pose, we warp the static regions of every candidate keyframe into the planned view using its estimated depth and camera pose, and measure the ratio of overlapping static pixels. This scalar score is used to rank all keyframes, and we retain the top-n views per timestep. The corresponding RGB values and diffusion features are bilinearly sampled from the source frame and warped into the conditioning tensor. Pixels that receive no valid projection remain black and are marked as holes; these unwarped regions act as an inpainting mask for the diffusion model. Importantly, the warped content is used as a soft spatial guide rather than a hard constraint, allowing the diffusion model to fill holes and refine details while still respecting the geometric layout where reliable projections exist. This ensures that the diffusion model retains the flexibility to generate dynamic details on top of this condition while the final output naturally blends geometric consistency with realistic temporal dynamics. This design is especially important in our framework, where we aim to generate scene-consistent future videos based on these spatial conditions. An example of our framework generating new dynamics on top of these spatial conditions are shown in Figure 8 of the Appendix.

---

### Author Response · Authors · 2025-11-26
**General Response**

We are significantly grateful for all the reviewers thoroughly reviewing our manuscript and providing valuable feedback, which can improve our work when addressed. We are also grateful that the reviewers recognize the strengths of our work, including the intuitiveness and effectiveness of the dual spatio-temporal conditioning strategy (3iY4, hP5a, xUs1) , the soundness of the static-only 3D scene memory and masking pipeline (3iY4, iskH, xUs1) , the clear motivation and novelty of the problem (hP5a, xUs1) , the clarity of the writing (iskH) , and the favorable performance demonstrated through extensive experiments and ablations (3iY4, hP5a, iskH).

Encouraged by the valuable feedbacks from the reviewers, we have conducted extra experiments which can verify the effectiveness of our approach. The revised pdf includes:

- Newly added teaser **Figure 1** for easier understanding of the pipeline of our framework.
- Updated **Figure 2** for improved clarity.
- Newly added an overview of our architecture in **Figure 3**.
- Additional qualitative videos in the **supplementary materials.**
- Additional experiments of applying our framework to scene-consistent camera-controllable ***long*** video generation via autoregressive generation in **Section A, Table 5, and Figure 14**.
- Comprehensive ablation of the dynamic masking pipeline in **Section A, Table 6**.
- Additional experiments of evaluating our framework with a single frame as a condition, making a fairer comparison with previous I2V camera-controllable video generation works in **Section A, Table 7**.
- Addition experiments of ablating the number of frames selected via spatial adjacency in **Section A, Table 8**.
- Additional experiments of replacing MegaSAM with DepthAnything v3 to showcase the flexibility and versatility of our framework in **Section A, Table 9 and Table 10**.
- Added a dedicated section of discussion with similar concurrent and prior works in **Section C**.
- Discussion of similarity between our proposed spatial prompts and existing video inpainting models in **Section C, Figure 8**.
- Additional qualitative results in **Figure 9, Figure 10, Figure 11, Figure 12, and Figure 13.**

We believe that thanks to the constructive feedback from the reviewers, our revised manuscript better highlights the strength of our proposed framework. We have highlighted modified text in ***blue*** where updated figures and tables are also emphasized with a ***blue bounding box***. We would be grateful if the ACs and reviewers could kindly review these updates.

---

### Author Response · Authors · 2025-12-01
**Summary of our work and discussions**

Dear ACs and reviewers,

We thank the ACs and reviewers for taking the time to thoroughly review our manuscript and providing insightful suggestions and feedback that helped us improve of our work. We would like to provide a concise summary of our work’s contributions, the reviewers' suggestions and concerns, and how we have resolved the concerns raised during the rebuttal period.

**1. Core Contribution of our work**

We propose **3DScenePrompt**, a framework for **scene-consistent camera-controllable video generation**.

- **Key Innovation:** We introduce a **dual spatio-temporal conditioning** strategy. Unlike prior methods that rely solely on temporal context (leading to a limited window size), we construct a **static-only 3D scene memory** (via SLAM and a novel dynamic masking pipeline). This allows us to project geometrically consistent "spatial prompts" from the input video to guide scene-consistent and camera-controllable future generation.
- During the discussion period, we have added additional experiments that better show the benefits of incorporating each of our conditioning, spatial conditioning for scene-consistency and precise camera-controllability, and temporal conditioning for better motion smoothness and coherence.

**2. Summary of Discussion**

We have addressed major concerns through additional experiments and text revisions.

- **Fairness of Comparison ($w=1$ Baseline):**
    - *Concern:* Reviewers 3iY4 and xUs1 noted that using 9 input frames ($w=9$) as temporal context gave us an advantage over existing I2V camera-controllable video generation works (CameraCtrl, MotionCtrl, FloVD).
    - *Resolution:* We clarified that our comparison with I2V works is due to these works being the state-of-the-art camera-controllable video generation works, with the codes being open-sourced. To better isolate the benefits of providing temporal frames, which is mainly for better motion coherency, we added a **single-frame ($w=1$) baseline** (Table 2 & 7). Our method outperforms existing baselines in camera controllability even with $w=1$, confirming the efficacy of our spatial prompts. This additional experiment was acknowledged by Reviewer xUs1.

- **Ablation of Spatial Prompts ($n=0$ Baseline):**
    - *Concern:* Reviewer 3iY4, hP5a, xUs1 requested verification that improvements aren't just from using more temporal frames.
    - *Resolution:* We added a baseline **without spatial prompts ($n=0$)** (Table 8). Results show a significant drop in both scene-consistency and camera-controllable metrics, proving the effectiveness of incorporating **3D spatial memory.** This additional experiment was acknowledged by Reviewer hP5a, xUs1.

- **Long-Video Generation & Generality:**
    - *Concern:* Reviewer iskH asked about long-video generation application.
    - *Resolution:* Although long-video generation is not our primary task of interest, we conduct additional experiments by autoregressively generating videos by iteratively applying our framework. This verifies the potential of applying our framework to long-video generations. We added **long-video generation experiments** on the **DAVIS dataset** (Table 5), outperforming previous I2V approaches when applying a similar autoregressive approach. This additional experiment was acknowledged by Reviewer iskH and xUs1.

- **End-to-End Computation Cost:**
    - *Concern:* Reviewer 3iY4 asked about the end-to-end computation cost.
    - *Resolution:* We have analyzed the end-to-end computation costs of our framework. In addition, as we introduce a versatile and flexible framework composed of a off-the-shelf SLAM module and pretrained DiT, we additionally demonstrated that we can easily replace MegaSAM with a more recently released DepthAnything v3 at inference, which reduces the overall inference time significantly while also improving performance (Table 9, Table 10). This additional experiment was acknowledged by Reviewer iskH and xUs1.

- **Comparisons with Concurrent Work:**
    - *Concern:* Reviewers asked about concurrent works (e.g., *CameraCtrl2*, *SPMem*).
    - *Resolution:* We first clarified that these works are concurrent works under the ICLR policy, tackle different tasks (e.g., lack of explicit scene consistency), or do not have their code publicly available. We added a dedicated discussion in Appendix C. This clarification was acknowledged by Reviewer iskH.

We believe the revised manuscript and additional experiments strongly validate the effectiveness of 3DScenePrompt.

---

### Meta-Review · Area_Chair_KcLR · 2026-01-07

**Summary:**

There are several major concerns raised:
- Robustness of SLAM model. The method relies on a explicit 3D scene memory represented reconstructed from SLAM. The reliance on the SLAM system can leads unsatisfactory results due to the limitation of SLAM on dynamic scenes or unseen views.
- Fairness. The method requires a dependence on the last 9 frames can compares to a subset of I2V models, which leads to unfairness in terms of the amount of input information.
- Large view change. A major concern on the qualitative experiments is whether the algorithm performs well for large camera motion (swing away and back).

**Reviewer Concerns:**

The fairness issue is addressed. However, the presented supplemental videos do not demonstrate a strong robustness for large view change setting. This also links to the robustness issue of the SLAM model.

**Reviewer Scores:**

Reviewer 3iY4 is likely to raise the score to 6.
Reviewer iskH is unlikely to change the score.

---

### Decision · Program_Chairs · 2026-01-26

Accept (Poster)